# Accelerated Dual Method for Distributed Optimization:
# An Inexact-Gradient View of Local Updates

Junchi Yang [* 1]  Ziyang Zeng [* 2]  Linxuan Pan [1]  Murat Yildirim [3]  Feng Qiu [4]

## Abstract

In distributed machine learning, efficiently training across multiple agents with heterogeneous data distributions remains a central challenge. We address the problem of stochastic, strongly convex distributed optimization by applying accelerated gradient ascent to the dual variables and multi-step stochastic gradient descent (SGD) to the primal variables in the Lagrangian formulation. This approach naturally enables local computation, as the inner SGD loops require no inter-agent communication. We prove that the method converges for any number of local updates, attaining the optimal communication complexity when local computation is sufficient. Our analysis builds on an inexact accelerated gradient framework, where the partial gradient of the Lagrangian with respect to the dual variables is treated as an inexact gradient of the dual function. A notable byproduct of this framework is an algorithm that achieves optimal reproducibility guarantees under biased gradient estimates.

## 1. Introduction

We consider unconstrained distributed optimization problems in the following formulation:

$$\min_{x \in \mathbb{R}^n} F(x) := \frac{1}{M} \sum_{m=1}^{M} F_m(x). \tag{1}$$

Here, $M$ clients, each with access to the gradient of its respective local function $F_m$, collaborate to minimize the aggregate of their objective functions. Such problems have

---

[*]Equal contribution [1]School of Data Science, The Chinese University of Hong Kong, Shenzhen [2]National University of Singapore [3]Industrial and Systems Engineering, Wayne State University [4]Northwestern University. Correspondence to: Junchi Yang <yangjunchi@cuhk.edu.cn>.

*Proceedings of the 43$^{rd}$ International Conference on Machine Learning*, Seoul, South Korea. PMLR 306, 2026. Copyright 2026 by the author(s).

garnered significant attention due to their relevance in various fields, including machine learning (Xing et al., 2016), signal processing (Rabbat & Nowak, 2004), power systems (Bidram et al., 2014), and control (Nedić & Liu, 2018). In particular in Federated Learning (Konečnỳ et al., 2016; McMahan et al., 2017), this form is prevalent as each agent possesses unique local data, so the local function is represented by $F_m(x) = \mathbb{E}_{\xi \sim P_m} f(x; \xi)$, where $P_m$ denotes the data distribution for the $m$-th agent. This approach allows all agents to train a global machine learning model represented by weight $x$, while ensuring that no local data are exchanged directly.

The trend toward larger machine learning models and training with more extensive datasets has introduced new challenges in distributed training. The transmission of gradient between clients incurs substantial communication overhead due to bandwidth constraints and the need for privacy-preserving protocols (Konecnỳ et al., 2016). Furthermore, when each client has a large pool of local data, estimates of local gradient $\nabla F_m$ are usually noisy. These factors challenge traditional algorithms that perform communication after each local computation of exact gradients (Nedić & Olshevsky, 2014; Shi et al., 2015; Nedic et al., 2017; Yuan et al., 2018; Tang et al., 2018). As a remedy, a popular method known as FedAvg (McMahan et al., 2017), or local SGD (Koloskova et al., 2020), performs multiple iterations of Stochastic Gradient Descent (SGD) on local objectives $F_m$ and then communicates the local model with its neighbors. However, this algorithm suffers from a suboptimal convergence rate and is hindered by heterogeneity, when local functions or data can vary significantly between clients (Zhao et al., 2018; Karimireddy et al., 2020).

In the deterministic setting, many algorithms are robust to heterogeneity, to list a few, Gradient Tracking (Nedic et al., 2017; Di Lorenzo & Scutari, 2016), Exact Diffusion (Yuan et al., 2018), D$^2$ (Tang et al., 2018). They can be extended to the stochastic setting (Koloskova et al., 2021; Pu & Nedi, 2018). Because clients communicate in each iteration, the communication complexity is of the same order as sample complexity, which is much worse than the lower communication complexity bounds (Scaman et al., 2017; 2019), unless an impractically large batch size of $\mathcal{O}(\epsilon^{-2})$ is

adopted by each client. Another way is to use *local steps*, i.e., communicate once after multiple local gradient updates, reducing the communication frequency. Exact Diffusion and Gradient Tracking (GT), combined with local steps, attain improved communication complexities even without minibatch (Alghunaim, 2024; Liu et al., 2024). However, these methods still cannot achieve the optimal complexity even when sufficient local steps are made. See Table 1.

Another family of deterministic algorithms attains optimal communication complexities by running accelerated gradient ascent (AGA) in the Lagrangian dual function after reformulating the distributed problem (1) as a linearly constrained problem (Scaman et al., 2017; Uribe et al., 2020). However, they rely on computing the gradients of the dual function exactly or $\mathcal{O}(\epsilon^2)$-accurately. If stochastic gradient methods are used in the primal variable to approximate the dual gradients, this entails $\mathcal{O}(\epsilon^{-2})$ local steps. In practice, the number of local steps is often limited by communication schedules and client computation capacity. By contrast, algorithms such as Local Exact Diffusion (LED) (Alghunaim, 2024) and Local Gradient Tracking (Liu et al., 2024) converge with an arbitrary number of local steps. This raises the natural question of whether AGA in the dual can also converge under arbitrary local steps.

We are interested in the following research question:

> *Does the accelerated dual method converge for an arbitrary number of local steps? Furthermore, can it attain optimal communication complexity when the number of local computations is below $\mathcal{O}(\epsilon^{-2})$ in the stochastic setting?*

### 1.1. Our Contributions

We consider a primal–dual algorithm for the Lagrangian reformulation of problem (1). The method performs multiple stochastic gradient descent iterations on the primal variables as an inner loop, followed by an accelerated gradient ascent step on the dual variables. We refer to this approach as *Local Accelerated Dual Ascent* (Local ADA). Since the inner SGD loop requires no inter-agent communication, Local ADA naturally supports local updates.

Local ADA can be viewed as a variant of the algorithms in (Scaman et al., 2017; Uribe et al., 2020), with $K$ local steps to approximately solve the conjugate function. We prove convergence for *any* number of local steps in the stochastic strongly convex setting. Moreover, when the number of local steps is $\widetilde{\Theta}(\epsilon^{-1})$, the method achieves the near-optimal communication complexity $\mathcal{O}(\kappa^{1/2}p^{-1/2})$, where $\kappa$ is the condition number and $p$ is the mixing parameter. Table 1 compares Local ADA with existing algorithms in scenarios where ample local data is available.

A key technical challenge lies in establishing convergence

for small numbers of local steps: in this regime, the partial gradient of the Lagrangian with respect to the dual variables is a highly inexact estimate of the gradient of the dual function. Since accelerated methods are notoriously sensitive to gradient inexactness (Attia & Koren, 2021), standard analyses do not apply. We develop a new analysis framework for accelerated gradient methods with inexact gradients, showing how the choice of momentum and step size can trade off contraction rate against bias reduction. Specifically, when local steps are few, smaller momentum and step size reduce the bias at the cost of a slower contraction rate.

An important byproduct of this framework is filling a gap left in (Ahn et al., 2022; Zhang et al., 2024): we provide the first algorithm that simultaneously achieves near-optimal convergence rates and reproducibility guarantees under biased gradient estimates.

### 1.2. Related Work

**Distributed optimization.** There has been a history of decentralized optimization (Tsitsiklis, 1984). The lower bound for communication complexities characterized by $p \triangleq 1 - \sigma_2$, where $\sigma_2$ is the second largest singular value of the weight matrix $W$ of the underlying graph, are discovered in strongly convex (Scaman et al., 2017), convex (Scaman et al., 2019) and nonconvex settings (Lu & De Sa, 2021). The matching algorithms have also been provided for strongly convex (Scaman et al., 2017; Uribe et al., 2020; Li et al., 2020; Li & Lin, 2020; Song et al., 2023), convex (Li et al., 2020; Uribe et al., 2020; Scaman et al., 2019; Li & Lin, 2020; Xu et al., 2020) and nonconvex settings (Lu & De Sa, 2021). However, they all have certain limitations: (a) most of them are restricted either to the deterministic setting (Li & Lin, 2020; Li et al., 2020; Song et al., 2023; Kovalev et al., 2020) or to the finite-sum local objectives (Shen et al., 2018; Mokhtari & Ribeiro, 2016); (b) some of them rely on Chebyshev acceleration that needs $\mathcal{O}(p^{-\frac{1}{2}})$ consensus steps in each communication rounds (Kovalev et al., 2020; Xu et al., 2020), which is burdensome when communication is a bottleneck; (c) being optimal only if large batch size is taken (Lu & De Sa, 2021). Several deterministic algorithms are extended to stochastic settings, with (Liu et al., 2024; Alghunaim, 2024) or without local steps (Koloskova et al., 2021), but they fail to attain the optimal communication complexities.

**(Primal)-Dual algorithms.** When the consensus of distributed optimization is written as a constraint, it is natural to consider primal-dual algorithms for solving the (augmented) Lagrangian. The Alternating Direction Method of Multipliers (ADMM) is a popular technique (Boyd et al., 2011; Wei & Ozdaglar, 2013; Makhdoumi & Ozdaglar, 2017). Several works have suggested using linearized primal subproblems (Chang et al., 2014; Hong & Chang, 2017; Ling

*Table 1.* Communication complexity of algorithms when sufficient local samples are available for minibatches or local steps. The column *Local Steps* indicates whether the algorithm performs local updates, while *Arbitrary K* indicates whether convergence is guaranteed for any number of local steps. Here, $\kappa$ denotes the condition number, and $p \triangleq 1 - \sigma_2$, where $\sigma_2$ is the second largest singular value of the network weight matrix. $M$ is the number of clients. $C_0$ in D-MSAG is defined in Fallah et al. (2022). Stochastic GT (Koloskova et al., 2021) involves another network-dependent constant $c \geq 1 - \sigma_2$. Local-DSGD assumes that heterogeneity is bounded by $\varsigma$.

| Algorithms | Communication Complexity | Local Steps | Arbitrary $K$ |
|---|---|---|---|
| Local-DSGD (Koloskova et al., 2020) | $\widetilde{\mathcal{O}}(\kappa p^{-1} + \sqrt{\kappa}\varsigma p^{-1}\epsilon^{-\frac{1}{2}})$ | Yes | Yes |
| D-MSAG (Fallah et al., 2022) | $\widetilde{\mathcal{O}}(\kappa^{\frac{1}{2}}p^{-\frac{1}{2}} + C_0\sqrt{\kappa}p^{-\frac{1}{2}}(M\epsilon)^{-\frac{1}{4}})$ | No | N/A |
| Stochastic GT (Koloskova et al., 2021) | $\widetilde{\mathcal{O}}(\kappa p^{-1}c^{-1})$ | No | N/A |
| LED (Alghunaim, 2024) | $\widetilde{\mathcal{O}}(\kappa^2 p^{-1})$ | Yes | Yes |
| Distributed FGM (Uribe et al., 2020) | $\widetilde{\mathcal{O}}(\kappa^{\frac{1}{2}}p^{-\frac{1}{2}})$ | Yes | No |
| Local ADA | $\widetilde{\mathcal{O}}(\kappa^{\frac{1}{2}}p^{-\frac{1}{2}})$ | Yes | Yes |
| Lower bound (Scaman et al., 2017) | $\widetilde{\Omega}(\kappa^{\frac{1}{2}}p^{-\frac{1}{2}})$ | N/A | N/A |

et al., 2015; Aybat et al., 2017), with a closed form for the primal update that involves a one-step (stochastic) Gradient Descent in the primal. Many algorithms, including EX-TRA (Shi et al., 2015; Mokhtari & Ribeiro, 2016; Li & Lin, 2020) and DIGing (Nedic et al., 2017), can be interpreted or motivated by the Gradient Descent Ascent algorithm on the augmented Lagrangian. The most closely related works to ours are (Terelius et al., 2011; Ghadimi et al., 2011; Scaman et al., 2017; Uribe et al., 2020), which executed (accelerated) dual ascent in the dual problem. However, they only consider deterministic settings and need to compute the exact or $\mathcal{O}(\epsilon^{-2})$-accurate dual gradients. Related variants include block-coordinate gradient descent on the dual objective (Necoara et al., 2017), dual proximal coordinate methods for finite-sum local functions (Hendrikx et al., 2019a), and asynchronous local-update schemes (Hendrikx et al., 2019b).

**Inexact gradient.** A long line of works investigate first-order algorithms when only an inexct gradient or subgradient is available (Devolder et al., 2013; 2014; d'Aspremont, 2008; Baes, 2009; Schmidt et al., 2011; Millán & Machado, 2019; Nedic & Bertsekas, 2001; Luo & Tseng, 1993). These works consider different notions of oracle error, including an accuracy level for first-order approximation and explicit bounds on the gradient error. In particular, Schmidt et al. (2011) analyze proximal-gradient methods with errors in both the gradient evaluation and the proximal step, where the gradient error is measured by the norm of the difference between the approximate and exact gradients. Ahn et al. (2022) introduce a notion of reproducibility as the deviation in outputs of independent runs of the algorithms when the gradient is $\delta$-inxact. They show Gradient Descent (GD) has an optimal reproducibility of $\mathcal{O}(\delta^2/\epsilon^2)$ but a suboptimal complexity $\mathcal{O}(\epsilon^{-1})$ in convex optimization. Zhang

et al. (2024) provide an algorithm with near-optimal $\widetilde{\mathcal{O}}(\epsilon^{-\frac{1}{2}})$ complexity but $\mathcal{O}(\delta^2/\epsilon^{2.5})$ reproducibility. We achieve both near-optimal complexity and reproducibility in this paper.

## 2. Decentralized Optimization

**Notations** By default, we use the Frobenius norm and inner product $\langle A, B \rangle = Tr(AB^\top)$ for matrices, and the Euclidean norm for vectors. The symbol $\mathbf{1}$ denotes a column vector consisting of all ones with an appropriate dimension.

In the decentralized setting, we study the problem in (1), where each client communicates only with its immediate neighbors in a network, without a central coordinator. The network is represented by a weight matrix $W \in \mathbb{R}^{M \times M}$, which satisfies the following assumptions:

**Assumption 2.1.** (1) $W = W^\top, I \succeq W \succeq -I$ and $W\mathbf{1} = \mathbf{1}$. (2) $\sigma_2 < 1$ where $\sigma_2$ is the second largest singular value of $W$. We denote $p \triangleq 1 - \sigma_2$.

Since $\sigma_2$ of $W$ may approach 1, our analysis will preserves the dependence on $(1 - p)^{-1}$. For each client $m$, the set of neighbors is denoted as $\mathcal{N}_m = \{j \neq m : W_{m,j} \neq 0\}$. Following Shi et al. (2015); Li et al. (2020), we introduce $U = (I - W)^{\frac{1}{2}} \in \mathbb{R}^{M \times M}$, which allows us to reformulate the problem as the constrained optimization:

$$\min_{X \in \mathbb{R}^{M \times n}} H(X), \quad \text{subject to} \quad UX = 0, \quad (2)$$

where $X = (x^1 \dots x^M)^\top \in \mathbb{R}^{M \times n}$ and $H(X) = \frac{1}{M}\sum_{i=1}^{M} F_m(x^m)$.

We aim to find a solution $X = (x^1, \dots, x^M)$ such that the average $\bar{x} = \frac{1}{M}\sum_{m=1}^{M} x^m$ is $\epsilon$-optimal for (1): $\mathbb{E}[f(\bar{x}) - f^*] \leq \epsilon$, where the expectation is taken over randomness in the algorithm and $f^*$ is the optimal value.

In addition, we provide a bound for consensus violation $\frac{1}{M} \sum_{m=1}^{M} \mathbb{E}\|x^m - \bar{x}\|^2 \leq \mathcal{O}(poly(\epsilon))$ in the proof. If a stronger consensus is required, one can apply an average consensus algorithm (Olfati-Saber et al., 2007; Xiao & Boyd, 2004) or its accelerated variant (Liu & Morse, 2011) after the main procedure. In particular, the accelerated algorithm achieves $\epsilon'$-consensus in only $\mathcal{O}(p^{-1/2} \log \frac{1}{\epsilon'})$ rounds of communications.

We focus on the strongly convex setting and adopt the following assumptions on the local objective functions:

**Assumption 2.2.** For each client $m$,
(1) $F_m$ is $L$-Lipsthiz smooth, i.e.,

$$\|\nabla F_m(x) - \nabla F_m(y)\| \leq L\|x - y\|, \quad \forall x, y.$$

(2) $F_m$ is $\mu$-strongly convex, i.e.,

$$F_m(y) \geq F_m(x) + \langle \nabla F_m(x), y - x \rangle + \frac{\mu}{2}\|y - x\|^2, \quad \forall x, y.$$

(3) Each client has access to the unbiased gradient estimate $g_m(x; \xi)$ such that $\mathbb{E}_\xi \, g_m(x; \xi) = \nabla F_m(x)$ and $\mathbb{E}_\xi \|g_m(x; \xi) - \nabla F_m(x)\|^2 \leq \sigma^2$.

We denote by $\kappa = L/\mu$ the condition number of the local objective functions.

### 2.1. Local Accelerated Gradient Ascent

Introducing dual variable $\lambda = (\lambda^1 \ldots \lambda^M)^\top \in \mathbb{R}^{M \times n}$, the Lagrangian of the constrained problem (2) is

$$\min_{X \in \mathbb{R}^{M \times n}} \max_{\lambda \in \mathbb{R}^{M \times n}} \mathcal{L}(X, \lambda) = H(X) + \langle \lambda, UX \rangle. \quad (3)$$

This formula is widely used in the literature (Li & Lin, 2020; Lan et al., 2020; Xu et al., 2020). We denote by $\widetilde{\nabla} H(X) = \frac{1}{M}(g_1(x^1; \xi^1) \cdots g_M(x^M; \xi^M))^\top$, a stochastic estimate of $\nabla H(X)$. Our algorithm applies stochastic gradient descent (SGD) on the primal variable $X$ for $K$ inner iterations, while updating the dual variable $\lambda$ using accelerated gradient ascent (AGA) (Nesterov, 2013):

$$X_{t,k+1} = X_{t,k} - \tau_2 \left[\widetilde{\nabla} H(X_{t,k}) + U\widetilde{\lambda}_t\right], \ k = 1, \ldots, K$$

$$\lambda_{t+1} = \widetilde{\lambda}_t + \tau_1 U X_{t,K} \quad (4)$$

$$\widetilde{\lambda}_{t+1} = \lambda_{t+1} + \beta(\lambda_{t+1} - \lambda_t),$$

Defining $\zeta = U\lambda$ and pre-multiplying the dual update by $U$, we obtain the equivalent updates:

$$X_{t,k+1} = X_{t,k} - \tau_2 \left[\widetilde{\nabla} H(X_{t,k}) + \widetilde{\zeta}_t\right], \ k = 1, \ldots, K$$

$$\zeta_{t+1} = \widetilde{\zeta}_t + \tau_1(I - W)X_{t,K}, \quad (5)$$

$$\widetilde{\zeta}_{t+1} = \zeta_{t+1} + \beta(\zeta_{t+1} - \zeta_t).$$

An important observation is that the update of $X$ in the inner loops does not require communication. We therefore refer to this as Local Accelerated Dual Ascent (Local

---

**Algorithm 1** Local Accelerated Dual Ascent (Local ADA)

1: **Input:** $\{x_0^m\}_{m=1}^M$, $\{\zeta_0^m\}_{m=1}^M$, stepsizes $\tau_1, \tau_2$, momentum $\beta$, inner steps $K$, outer steps $T$
2: **Initialize:** $\widetilde{\zeta}_0^m = \zeta_0^m$ for all $m \in [M]$
3: **for** $t = 0, 1, \ldots, T - 1$ **do**
4:    **for** $k = 0, 1, \ldots, K - 1$ **do**
5:       **for all** $m \in [M]$ **in parallel do**
6:          sample $\{\xi_k^m\}_k$
7:          $x_{t,k+1}^m = x_{t,k}^m - \tau_2 \left[\frac{1}{M} g_m(x_{t,k}^m; \xi_k^m) + \widetilde{\zeta}_t^m\right]$
8:       **end for**
9:    **end for**
10:    **for all** $m \in [M]$ **in parallel do**
11:       $x_{t+1,0}^m = x_{t,K}^m$
12:       $\zeta_{t+1}^m = \widetilde{\zeta}_t^m + \tau_1\left(x_{t+1,0}^m - \sum_{j \in \mathcal{N}_m} W_{m,j} x_{t+1,0}^j\right)$
13:       $\widetilde{\zeta}_{t+1}^m = \zeta_{t+1}^m + \beta(\zeta_{t+1}^m - \zeta_t^m)$
14:    **end for**
15: **end for**
16: **Output:** $\{x_{t,k}^m\}_{m=1}^M$

---

ADA), presented in Algorithm 1. Substituting $X_{t,K}$ with $X_*(\zeta_t) \triangleq \arg\min_X H(X) + \langle \zeta_t, X \rangle$ in the update of $\zeta_{t+1}$ reduces the method to the accelerated dual method by Scaman et al. (2017). When the dual gradient $X^*(\zeta)$ cannot be computed exactly, Uribe et al. (2020) employed accelerated gradient descent in the inner loop to obtain a high-accuracy approximation of order $\mathcal{O}(\epsilon^{-2})$. In practice, however, computing such a high-accuracy approximation is very costly in the stochastic setting and can be limited by local resources. In contrast, our method relies on a flexible number of $K$ steps of SGD in the inner loop.

**Comparison with other algorithms.** Setting the momentum parameter $\beta = 0$ in Local ADA yields a method closely related to *Local Exact Diffusion (LED)*, recently introduced by Alghunaim (2024). When the number of local steps is set to one, LED further reduces to *Exact Diffusion* (Yuan et al., 2018) and $D^2$ (Tang et al., 2018). We also describe a centralized variant in Algorithm 3. When $\beta = 0$, this variant closely resembles *Scaffnew* (Mishchenko et al., 2022), with the main difference that Scaffnew employs a random number of inner-loop iterations. In this case, the control variate used in Scaffnew to mitigate client drift can be naturally interpreted as the dual variable in our Lagrangian formulation.

### 2.2. Convergence

Based on its update rule (4), we observe that: $\lambda^t$ remains in $\text{span}(U) = \{U\hat{\lambda} : \hat{\lambda} \in \mathbb{R}^{M \times n}\}$, as long as the initial $\lambda^0 \in \text{Span}(U)$. Additionally, according to Lemma 3.1 in (Shi et al., 2015), an optimal $\lambda^*$ exists in $\text{Span}(U)$.

We now present an intermediate step for completeness. The idea has been discovered in (Scaman et al., 2017), but they assume the existence of the Hessian of the Fenchel conjugate of $f$, which can be met when $f$ is twice differentiable. When $A$ is an identity matrix or full-rank, it reduces to the classic results in (Rockafellar & Wets, 2009; Kakade et al., 2009).

**Proposition 2.3.** *Assume the function* $f : \mathbb{R}^{m \times n} \to \mathbb{R}$ *is* $\nu$-*strongly convex and* $\ell$-*smooth. Let* $A \in \mathbb{R}^{d \times m}$ *be a matrix, with* $\sigma_{\max}$ *representing its maximum singular values. Then, the function* $h(\lambda) = \min_x \{f(x) + \langle \lambda, Ax \rangle\}$ *is* $\sigma_{\max}^2/\nu$-*smooth. Also,* $h$ *is* $\sigma_{\min}^2/\ell$-*strongly concave in the subspace* $\mathrm{span}(A)$*, where* $\sigma_{\min}$ *is the smallest nonzero singular values of* $A$*. This means that with* $\lambda_1, \lambda_2 \in \mathrm{span}(A)$*, we have*

$$\langle \nabla h(\lambda_2) - \nabla h(\lambda_1), \lambda_1 - \lambda_2 \rangle \geq \frac{\sigma_{\min}^2}{\ell} \|\lambda_1 - \lambda_2\|^2.$$

According to this lemma, the dual function $\Psi(\lambda) \triangleq \min_X \mathcal{L}(X, \lambda)$ is $L_\Psi$-smooth and $\mu_\Psi$-strongly concave in $\mathrm{span}(U)$, where $L_\Psi = 2M/\mu$ and $\mu_\Psi = M(1 - \sigma_2)/L$. Hence its condition number is $2\kappa/(1 - \sigma_2)$, where $\kappa = L/\mu$ is the condition number of the local objective functions. Consequently, if the exact dual gradient $\nabla \Psi(\lambda)$ is available, accelerated gradient ascent converges linearly.

**Theorem 2.4.** *Under Assumptions 2.1 and 2.2, consider running Algorithm 1 with stepsizes* $\tau_1$, $\tau_2$ *and momentum parameter* $\beta > 0$.

(a) *When local steps* $K \leq \widetilde{\Theta}\big(\max\{c_0 \sigma^2/\epsilon, \ \kappa\}\big)$*, the number of communication rounds to achieve an* $\epsilon$-*accurate solution is* $\widetilde{\mathcal{O}}(c_1/(\epsilon K))$.

(b) *When* $K \geq \widetilde{\Theta}\big(\max\{c_0 \sigma^2/\epsilon, \ \kappa\}\big)$*, the number of communication rounds is* $\widetilde{\mathcal{O}}((\kappa/(1 - \sigma_2))^\alpha)$ *for any* $\frac{1}{2} \leq \alpha \leq 1$*, provided that the momentum parameter is chosen as* $\beta = 1 - \Theta\big((\mu\tau_1)^{1-\alpha}\big) > 0$.

*Here,* $c_0$ *and* $c_1$ *are constants depending on the problem parameters and* $\alpha$*, and the notations* $\widetilde{\mathcal{O}}(\cdot)$ *and* $\widetilde{\Theta}(\cdot)$ *hide logarithmic factors.*

A more explicit statement is provided in Theorem C.2. The theorem shows that Local ADA converges for any number of local SGD steps. In contrast to Uribe et al. (2020), our method does not require the inner problem to be solved exactly or to $\mathcal{O}(\epsilon^2)$ accuracy. When the number of local steps $K$ is moderate, the total sample complexity, given by the product of the number of communication rounds and $K$, is $\widetilde{\mathcal{O}}(c_0\epsilon^{-1})$, which is near-optimal in the dependence of $\epsilon$.

When sufficiently many local steps are used, namely $K = \widetilde{\Theta}(c_0\sigma^2/\epsilon)$ in the stochastic setting and $K = \widetilde{\Theta}(\kappa)$ in the deterministic setting ($\sigma = 0$), Local ADA exhibits nearly linear outer-loop convergence and achieves communication complexity $\widetilde{\mathcal{O}}((\kappa/(1 - \sigma_2))^\alpha)$ for any $\alpha \in [1/2, 1]$. Thus,

using a larger momentum parameter improves the communication complexity. In particular, taking $\alpha = 1/2$ yields the near-optimal rate $\widetilde{\mathcal{O}}(\sqrt{\kappa/(1 - \sigma_2)})$, which matches the lower bound of (Scaman et al., 2017) up to logarithmic factors.

### 2.3. Extension to Non-strongly Convex Settings

Our main analysis focuses on the strongly convex setting. Nevertheless, the proposed dual method can also be used as an inner solver within an accelerated proximal point (Frostig et al., 2015; Monteiro & Svaiter, 2013) or Catalyst-type framework (Lin et al., 2018; Lan & Li, 2023). Such a framework has been adapted to decentralized optimization in (Li & Lin, 2020; Cao et al., 2026); see Algorithm 4.

At the $s$-th outer iteration, the framework constructs a regularized distributed subproblem with local objectives

$$F_m^s(x) = F_m(x) + \frac{\rho}{2}\|x - y_m^s\|^2,$$

where $\rho > 0$ is the proximal parameter and $y_m^s$ is the extrapolated point. For a suitable choice of $\rho$, each subproblem becomes strongly convex and can therefore be solved by Algorithm 1. We defer further details to Section D. Since this reduction introduces an additional outer loop and has been extensively studied in the literature, we do not focus on it in the main paper.

**Corollary 2.5.** *Suppose Assumptions 2.1 and 2.2(1) hold. Consider Algorithm 4, where each regularized subproblem is solved by Algorithm 1 with sufficiently many local steps. The communication complexity is* $\widetilde{\mathcal{O}}\left(\frac{1}{\sqrt{(1-\sigma_2)\epsilon}}\right)$ *for finding* $\mathbb{E}[F(\bar{x}) - F^\star] \leq \epsilon$ *in the convex case, and* $\widetilde{\mathcal{O}}\left(\frac{1}{\sqrt{1-\sigma_2}\,\epsilon^2}\right)$ *for finding* $\mathbb{E}\|\nabla F(\bar{x})\|^2 \leq \epsilon^2$ *in the smooth nonconvex case.*

These communication complexities match the known lower bounds for smooth convex and smooth nonconvex decentralized optimization up to logarithmic factors (Scaman et al., 2019; Lu & De Sa, 2021).

## 3. Analysis Framework

In our analysis, we work with the equivalent form (4) of Algorithm 1. In the update of $\lambda_{t+1}$, the partial gradient of the Lagrangian $U X_{t,K}$ can be viewed as an inexact approximation of the dual gradient $\nabla \Psi(\lambda) = U X^*(\lambda_{t+1})$ where $X^*(\lambda) = \arg\min_X H(X) + \langle \lambda, UX \rangle$. We first establish an inexact gradient analysis for the accelerated gradient method (Nesterov, 2013).

Consider the standard minimization problem

$$\min_{x \in \mathcal{X}} f(x)$$

where $\mathcal{X}$ is a convex and closed domain. Suppose we have access only to an inexact gradient oracle, as defined below.

**Definition 3.1.** The $\delta$ inexact gradient oracle for a function $f(x)$ is defined as: for any point $x$, the oracle returns $g(x)$ such that $\|\nabla f(x) - g(x)\| \leq \delta$.

We consider accelerated gradient descent (AGD) with an inexact oracle:

$$
\begin{aligned}
x_{t+1} &= \Pi_\mathcal{X} \left( y_t - \tau\, g(y_t) \right), \\
y_{t+1} &= x_{t+1} + \beta(x_{t+1} - x_t).
\end{aligned}
\quad \text{(Inexact-AGD)}
$$

**Theorem 3.2.** *Suppose that $f$ is $\mu$-strongly convex and $\ell$-smooth over a closed convex set $\mathcal{X}$, and that we have access to a $\delta$-inexact gradient oracle as in Definition 3.1. Starting from $y_0 = x_0$, and using a stepsize $\tau \leq \frac{1}{2\ell}$, Inexact-AGD with $\beta = 0$ satisfies, for any $\theta \in (0, \frac{\mu\tau}{2}]$,*

$$
\Delta_{t+1} \leq (1-\theta)\Delta_t + \left( \tau + \frac{\theta}{\mu} \right) \delta^2,
$$

*and for $\theta \in \left( \frac{\mu\tau}{2}, \sqrt{\frac{\mu\tau}{2}} \right]$, with $\beta = \frac{(2\theta - \mu\tau)(1-\theta)}{2\theta - \mu\tau\theta}$, it holds that*

$$
\Delta_{t+1} + \frac{2\theta^2 - \theta\mu\tau}{4\tau(1-\theta)} \|u_{t+1} - x^*\|^2
$$

$$
\leq (1-\theta)\left( \Delta_t + \frac{2\theta^2 - \theta\mu\tau}{4\tau(1-\theta)} \|u_t - x^*\|^2 \right) + \left( \frac{\tau}{2} + \frac{2\theta}{\mu} \right) \delta^2,
$$

*where $\Delta_t = f(x_t) - \min_{x \in \mathcal{X}} f(x)$ and $u_t = \frac{1}{\theta} x_t + \frac{1-\theta}{\theta} x_{t-1}$.*

The theorem characterizes the per-iteration convergence behavior: a potential function contracts at rate $1 - \theta$, up to a bias term determined by the inexactness $\delta$. The stepsize and momentum jointly create a tradeoff: faster contraction comes at the cost of a larger bias. In the unaccelerated case, i.e., $\theta = \frac{\mu\tau}{2}$, the error term decreases linearly with $\tau$. In the accelerated case, i.e., $\theta = \sqrt{\frac{\mu\tau}{2}}$, the error term scales as $\sqrt{\tau}$. Note that $\beta$ decreases with $\theta$ in this range. This tradeoff is central to our proof of Local ADA.

**Corollary 3.3.** *Under the assumptions of Theorem 3.2, if we choose $\tau = \frac{1}{2\ell}$ and $\beta = \frac{1 - \sqrt{\mu/(4\ell)}}{1 + \sqrt{\mu/(4\ell)}}$, then*

$$
\Delta_T \leq \left( 1 - \sqrt{\frac{\mu}{4\ell}} \right)^T \left( \Delta_0 + \frac{\mu}{4} D_0^2 \right) + \left( \frac{1}{2\sqrt{\ell\mu}} + \frac{1}{\mu} \right) \delta^2,
$$

*where $D_0 = \|x_0 - x^*\|$.*

This corollary shows that the algorithm converges at the optimal accelerated rate $\mathcal{O}\left( e^{-T/(2\sqrt{\kappa})} \right)$, up to a $\mathcal{O}(\delta^2/\mu)$-neighbourhood of the optimal value, where $\kappa = \ell/\mu$ is the condition number.

This improves over existing guarantees for inexact accelerated methods. Schmidt et al. (2011) study a setting in which the inexactness may vary across oracle calls. When the oracle inexactness is fixed at $\delta$, their result implies the same accelerated convergence rate as Corollary 3.3, but only up to an $\mathcal{O}(\delta^2\kappa/\mu)$-neighborhood of the optimum; see their Proposition 4. Devolder et al. (2013) introduced an alternative notion of inexact oracle, under which a $\delta$-inexact gradient in Definition 3.1 corresponds to a $\Theta(\delta^2/\mu)$-inexact gradient in their framework[1]. By their convergence result, the inexact accelerated gradient method converges to a neighborhood of size $\mathcal{O}(\delta^2\sqrt{\kappa}/\mu)$, which is also worse than our bound. Later, we will see that this improvement is the key to obtaining an optimal reproducibility algorithm in Section 3.2.

### 3.1. Proof Sketches for Theorems 2.4 and 3.2

We now outline the main ideas underlying the proofs of the two theorems, with the goal of providing intuition on how the proof of Theorem 2.4 relates to that of Theorem 3.2.

**Proof sketch for Theorem 3.2.** Using the smoothness and strong convexity of $f$, we first incorporate the $\delta$-inexact oracle into two fundamental one-step relations of $f(x_{t+1}) - f(x^*)$ and $f(x_{t+1}) - f(x_t)$. Following the classical analysis of AGD, we introduce an auxiliary sequence $u_t$ and a scalar $v = v(\theta, \beta)$ that depends on the contraction parameter $\theta$ and the momentum parameter $\beta$. We then carefully establish the following recursion:

$$
\Delta_{t+1} + \frac{\theta^2}{2\tau} \|u_{t+1} - x^*\|^2
$$

$$
\leq (1-\theta) \left[ \Delta_t + \frac{\theta^2 v}{2\tau(1-\theta)} \|u_{t+1} - x^*\|^2 \right]
$$

$$
+ \left[ \frac{\mu\theta}{4} - \frac{\theta^2(1-v)}{2\tau} \right] \|y_t - x^*\|^2
$$

$$
- \rho(v, \theta, \tau)\|y_t - x_t\|^2 + \left( \tau + \frac{\theta}{\mu} \right) \delta^2
$$

$$
- (1-\theta)(\nabla f(y_t) - g(y_t))^\top (x_t - y_t).
$$

where $\rho$ is a quantity that depends on $v$, $\theta$, and $\tau$. To ensure that the first two lines form a contraction for a potential function of the form $\Delta_t + c\|u_t - x^*\|^2$, we require $\frac{\theta^2}{2\tau} \geq \frac{\theta^2 v}{2\tau(1-\theta)}$. In addition, we require the coefficient of $\|y_t - x^*\|^2$ to be non-positive. Under these conditions, we obtain constraints relating $v$ and $\theta$, and we choose the parameters to maximize $\rho(v, \theta, \tau)$. Finally, by Young's inequality, the

---

[1] Devolder et al. (2013) define a $(\delta, \tilde{\ell}, \tilde{\mu})$-inexact oracle that, at a point $y$, returns $(\tilde{f}(y), \tilde{g}(y))$ such that for all $x$, $\frac{\tilde{\mu}}{2}\|x - y\|^2 \leq f(x) - \tilde{f}(y) - \langle \tilde{g}(y), x - y \rangle \leq \frac{\tilde{\ell}}{2}\|x - y\|^2 + \delta$. A $\delta$-inexact oracle in Definition 3.1 can be translated into a $\left( \frac{\delta^2}{\mu} + \frac{\delta^2}{2\ell}, \frac{\mu}{2}, 2\ell \right)$-inexact oracle. See their Section 2.3.

negative term $-\rho(v, \theta, \tau)\|y_t - x_t\|^2$ can be used to offset the last inner-product term, creating an additional error term involving $\|\nabla f(y_t) - g(y_t)\|^2 \le \delta^2$. The detailed proof can be found in Appendix B.

**Proof sketch for Theorem 2.4.** In Local ADA, the outer loop uses the inexact gradient $g_t = UX_{t,K}$ in place of $\nabla\Psi(\tilde{\lambda}_t)$, yielding the gradient error $\delta_t = \|\nabla\Psi(\tilde{\lambda}_t) - g_t\|$ at iteration $t$. We construct a potential $P_t$ contains three terms: (i) an outer-loop Lyapunov part $A_t \propto \Psi^* - \Psi(\lambda_{t+1}) + \|u_t - \lambda^*\|^2$, which is analogous to the inexact-AGD potential function. (ii) the inner-loop gap $B_t := \mathbb{E}\big[\mathcal{L}(X_{t+1}, \tilde{\lambda}_t) - \Psi(\tilde{\lambda}_t)\big]$, which is introduced to control the inexact dual gradient error $\delta_t$, and (iii) the consensus proxy $\|\lambda_{t+1} - \tilde{\lambda}_t\|^2$.

We briefly comment on how the terms $A_t$ and $B_t$ are bounded. (a) The term $A_t$ can be controlled using arguments motivated by Theorem 3.2, albeit with several important modifications. The inexactness can be bounded in terms of $B_t$. (b) The term $B_t$ captures the progress of the inner loop. Since the inner loop performs SGD, after $K$ iterations it can be bounded by $(1 - \tau_2\mu)^K\big[\mathcal{L}(X_t, \tilde{\lambda}_t) - \Psi(\tilde{\lambda}_t)\big]$, up to an additional noise term. To relate this bound to $B_{t-1}$, we further need to control the difference $\Psi(\tilde{\lambda}_t) - \Psi(\tilde{\lambda}_{t-1})$.

With these bounds, we establish the Lyapunov recursion

$$P_{t+1} \le q\,P_t + \mathrm{noise}(\tau_2, K), \qquad q \in (0, 1),$$

provided that a collection of parameter conditions is satisfied (see Lemma C.1). These conditions include requiring $\tau_1$ to be sufficiently small relative to $\tau_2$ when the number of local steps $K$ is small. The contraction rate $q$ takes the form $\max\{1 - \theta,\ r(\theta, \tau_1, \tau_2, K)\}$, where $r$ decreases as $\theta$, $\tau_2$, or $K$ increases. The noise term decreases when $\tau_2$ is small. When $K$ is small, to make the noise term small, $\tau_2$ (and consequently $\tau_1$) must be chosen small, which leads to a slower contraction rate $q$. In contrast, when $K$ is large, the term $r$ becomes small, and the contraction rate $q$ is dominated by $1 - \theta$, which depends on the momentum parameter $\beta$. The complete proof is deferred to Appendix C.

### 3.2. Reproducibility

We now shift our focus to an important byproduct of Corollary 3.3. Ahn et al. (2022) introduced a notion of reproducibility for optimization algorithms. For a convex and $\ell$-smooth optimization problem $\min_{x \in \mathcal{X}} f(x)$, reproducibility is measured by the distance between the outputs of two independent runs of the algorithm under certain sources of perturbation, such as inexact gradients. The formal definition is as follows.

**Definition 3.4.** The $(\epsilon, \delta)$-deviation, defined as $\|\hat{x} - \hat{x}'\|^2$, measures the reproducibility of an algorithm $\mathcal{A}$ with $\epsilon$-optimal solutions $\hat{x}$ and $\hat{x}'$, where $\hat{x}$ and $\hat{x}'$ are the outputs

---

**Algorithm 2** Reproducible Inexact-AGD

1: **Input:** regularization parameter $r > 0$, accuracy $\epsilon_r > 0$, initial point $x_0 \in \mathcal{X}$
2: **Initialize:** auxiliary problem

$$f_r(x) = f(x) + \tfrac{r}{2}\|x - x_0\|^2 \qquad (\star)$$

which is $r$-strongly convex and $(\ell + r)$-smooth
3: Apply Inexact-AGD to approximately solve

$$x_r = \arg\min_{x \in \mathcal{X}} f_r(x)$$

such that the optimality gap satisfies

$$f_r(x_r) - \min_{x \in \mathcal{X}} f_r(x) \le \epsilon_r$$

4: **Output:** $x_r$

---

of two independent runs of $\mathcal{A}$ given a $\delta$-inexact oracle in Definition 3.1.

The goal is to design an algorithm that not only achieves near-optimal solutions at a fast rate but also guarantees reproducible outputs. Following Zhang et al. (2024), we employ Inexact-AGD to solve a quadratically regularized auxiliary problem $(\star)$, presented in Algorithm 2. The resulting regularized problem is strongly convex and smooth, allowing our theoretical guarantees for Inexact-AGD to be directly applied. The following theorem presents the gradient complexity and reproducibility.

**Theorem 3.5.** *Assume $f$ is convex and $\ell$-smooth. Let $\|x_0 - x^*\| \le D$ for some optimal solution $x^*$, and suppose $0 < \epsilon \le \ell D^2$. Given a $\delta$-inexact gradient oracle, Algorithm 2 with $r = \epsilon/D^2$ and $\epsilon_r = 3\delta^2/r$ returns a point $x_r$ that is $(3\delta^2 D^2/\epsilon + \epsilon/2)$-optimal, with gradient complexity $\tilde{\mathcal{O}}\big(\sqrt{\ell D^2/\epsilon}\big)$. Moreover, the reproducibility is $\mathcal{O}(\delta^2/\epsilon^2)$.*

This theorem implies that, when $\delta \le \mathcal{O}(\epsilon)$, Algorithm 2 converges to an $\epsilon$-optimal solution with near-optimal complexity and reproducibility, matching the lower bounds of Nesterov (2013) and Ahn et al. (2022), respectively. This is due to we improve the analysis of Inexact-GDA in the size of bias term that stems from gradient inexactness over (Zhang et al., 2024) in Corollary 3.3. Achieving a reproducibility bound that matches the known lower bound further suggests that the inexact neighborhood size in Corollary 3.3 should be optimal in its dependence on the condition number $\kappa$.

Previously, Ahn et al. (2022) showed that gradient descent (GD) achieves optimal reproducibility of $\mathcal{O}(\delta^2/\epsilon^2)$, but with a suboptimal complexity of $\mathcal{O}(1/\epsilon)$ when $\delta \le \mathcal{O}(\epsilon)$. In contrast, Zhang et al. (2024) obtained a near-optimal complex-

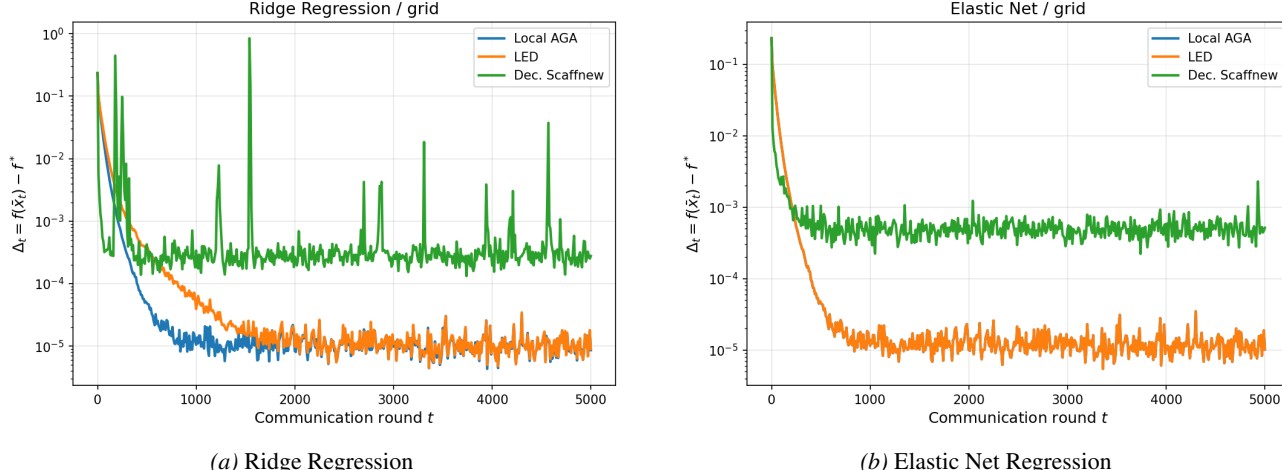

*(a)* Ridge Regression                 *(b)* Elastic Net Regression

*Figure 1.* Comparison of Local ADA, LED, and Decentralized Scaffnew for elastic net and ridge regression on the YearPredictionMSD dataset using a $4 \times 5$ grid.

ity of $\tilde{\mathcal{O}}(1/\sqrt{\epsilon})$, but only the reproducibility of $\mathcal{O}(\delta^2/\epsilon^{5/2})$, when $\delta \leq \mathcal{O}(\epsilon^{5/4})$. Both works conjectured that their reproducibility and complexity bounds cannot be improved simultaneously. We disprove this conjecture.

## 4. Numerical Experiments

We evaluate the performance of Local Accelerated Gradient Ascent (Local ADA) under decentralized and heterogeneous data distributions. We consider two types of objectives: regularized least-squares regression on the YearPrediction-MSD dataset and regularized logistic regression on the w8a dataset.

**YearPredictionMSD dataset.** We conduct two regression experiments on the YearPredictionMSD dataset (Bertin-Mahieux, 2011): ridge regression and elastic net regression. The dataset contains approximately 515K samples with 90 features. We use $M = 20$ clients connected by a $4 \times 5$ grid graph ($\delta \approx 0.086$), and adopt shard-based partitioning to induce data heterogeneity across clients.

**w8a dataset.** We further evaluate Local ADA on a regularized logistic-regression task using the w8a dataset (Chang & Lin, 2011). To create a non-IID client split, we apply a shard-style partition: we sort all examples by the value of one feature coordinate, split the sorted sequence into $M$ contiguous blocks of equal size, and assign one block to each client. As a result, different clients receive samples from different regions of the feature space, leading to heterogeneous local data distributions. In this experiment, we consider three decentralized communication graphs over $M = 30$ clients: a ring, a $5 \times 6$ grid, and an exponential graph, where each node connects to neighbors at distances $2^k$ modulo $M$. In terms of the spectral gap of the mixing matrix $W$, the

exponential graph is the most connected topology, followed by the $5 \times 6$ grid and then the ring.

### 4.1. Experiment Results

We compare Local ADA with LED (Alghunaim, 2024), Decentralized Scaffnew, and Local-DSGD (Koloskova et al., 2020), depending on the benchmark. In each experiment, Local ADA and LED use the same number of local computation steps. All other hyperparameters are tuned by grid search for each method.

Figure 1 presents the results on the YearPredictionMSD dataset. For both ridge and elastic net regression, Local ADA rapidly reduces the objective gap and attains a low final error comparable to LED. In particular, on ridge regression, Local ADA converges faster than LED in the early stage. In contrast, Decentralized Scaffnew stagnates at a noticeably higher objective gap in both settings and exhibits larger fluctuations on ridge regression. These results demonstrate that Local ADA is effective for large-scale heterogeneous regression problems.

Figure 2 reports the performance on the w8a logistic-regression task under different decentralized topologies. Across the exponential graph, $5 \times 6$ grid, and ring, Local ADA consistently achieves the best final objective gap among the compared methods. Although LED often decreases quickly during the initial phase, it tends to oscillate around a higher error level. Local ADA continues to improve over later communication rounds and eventually reaches a lower objective gap. In contrast, Local-DSGD converges substantially more slowly and remains at a much higher objective gap, confirming the benefit of acceleration in heterogeneous decentralized optimization.

Finally, Figure 3 examines the effect of the number of local

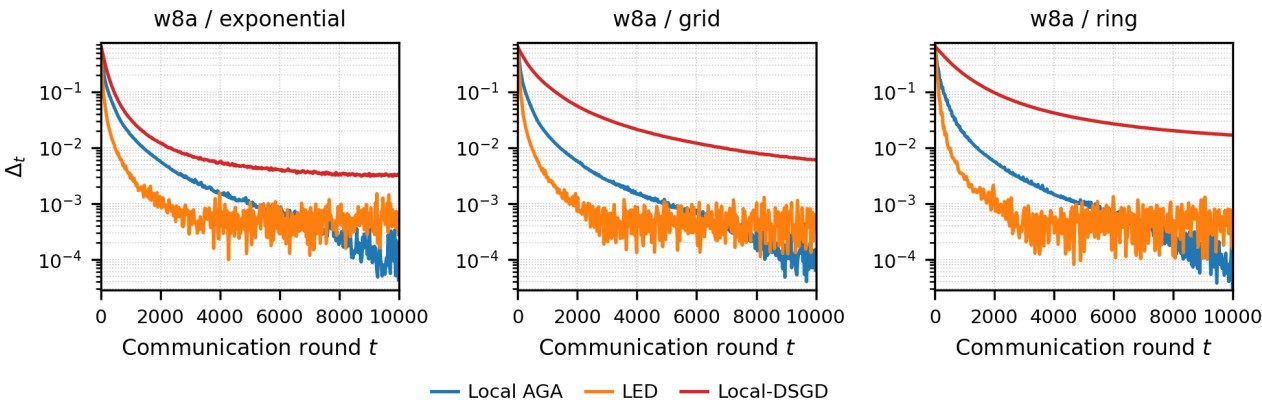

*Figure 2.* Comparison of Local ADA, LED, and Local-DSGD on the w8a logistic-regression task under three decentralized topologies: exponential graph, $5 \times 6$ grid, and ring. The objective gap $\Delta_t = f(\bar{x}_t) - f^*$ is plotted against communication rounds $t$, with each curve corresponding to the best-tuned hyperparameters.

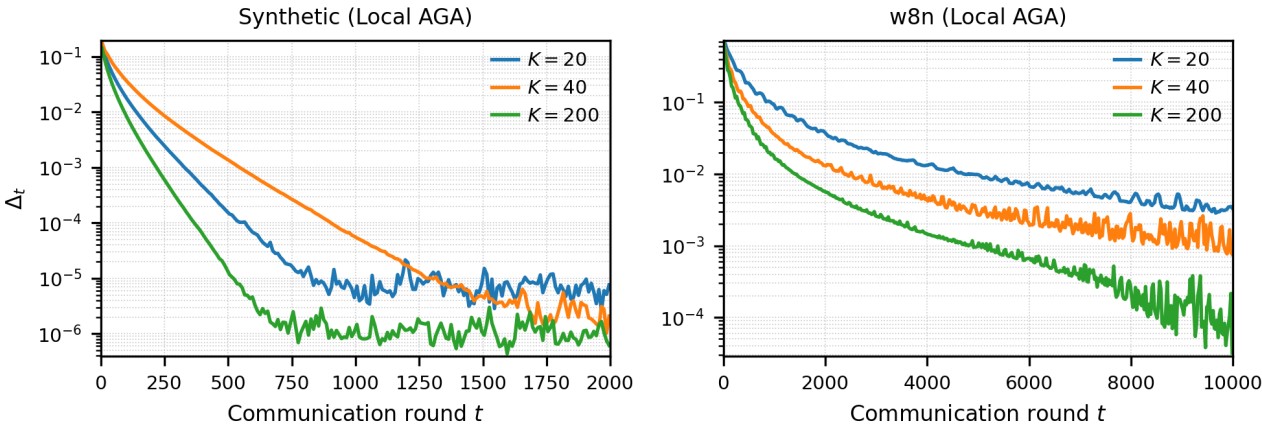

*Figure 3.* Effect of the number of local computation steps $K$ on Local ADA for logistic regression on the w8a and synthetic datasets. We plot the objective gap $\Delta_t = f(\bar{x}_t) - f^*$ versus communication rounds $t$ for $K \in \{20, 40, 200\}$.

computation steps $K$ on Local ADA for the w8a dataset and a synthetic dataset with 30 clients. Increasing $K$ from 20 to 200 consistently improves communication efficiency. In particular, $K = 200$ yields the fastest decrease and the lowest final objective gap among the tested choices. This confirms the main empirical advantage of Local ADA: by performing more local computation between communication rounds, it can significantly reduce the number of communication rounds required to reach high accuracy.

## 5. Conclusion

We studied accelerated dual methods for strongly convex stochastic decentralized optimization. In contrast to prior work, which typically requires solving the inner problem exactly or to $\mathcal{O}(\epsilon^2)$ accuracy in order to approximate the dual problem, we establish convergence guarantees for an arbitrary number of local steps. Moreover, we show that only

$\widetilde{\mathcal{O}}(\epsilon^{-1})$ local steps are sufficient to achieve near-optimal communication complexity $\widetilde{\mathcal{O}}(\kappa^{1/2} p^{-1/2})$.

From a technical perspective, our analysis builds on accelerated gradient descent with inexact gradient oracles. We uncover a fundamental trade-off between the convergence rate and the size of the inexact neighborhood by appropriately tuning the momentum parameter and stepsizes. In the decentralized setting, this trade-off manifests as a balance between the number of local steps $K$ and the convergence rate measured in communication rounds. As a byproduct, our refined inexact-AGD analysis yields an optimally reproducible algorithm for convex smooth optimization problems.

## Impact Statement

This paper presents work whose goal is to advance the field of Machine Learning. There are many potential societal

consequences of our work, none which we feel must be specifically highlighted here.

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

# A. Proofs for Preliminaries

**Proposition A.1.** *If each $F_m$ is $L$-smooth, i.e., $\|\nabla F_m(x) - \nabla F_m(\hat{x})\| \leq L\|x - \hat{x}\|$ for all $x$ and $\hat{x}$, then $H(X) = \frac{1}{M}\sum_{i=1}^{M} F_m(x_m)$ is $(L/M)$-smooth in $X$. If each $F_m$ is $\mu$-strongly convex, then $H(X)$ is $(\mu/M)$-strongly convex in $X$.*

*Proof.* When each $F_m$ is $L$-smooth,

$$
\|\nabla H(X) - \nabla H(\hat{X})\|^2 = \left\| \frac{1}{M}\begin{pmatrix} \nabla F_1(x_1)^\top \\ \nabla F_2(x_2)^\top \\ \vdots \\ \nabla F_M(x_M)^\top \end{pmatrix} - \frac{1}{M}\begin{pmatrix} \nabla F_1(\hat{x}_1)^\top \\ \nabla F_2(\hat{x}_2)^\top \\ \vdots \\ \nabla F_M(\hat{x}_M)^\top \end{pmatrix} \right\|^2
$$

$$
= \frac{1}{M^2}\sum_m \|\nabla F_m(x_m) - \nabla F_m(\hat{x}_m)\|^2
$$

$$
\leq \frac{1}{M^2}\sum_m L^2\|x_m - \hat{x}_m\|^2 = \frac{L^2}{M^2}\|X - \hat{X}\|^2.
$$

Therefore, $H$ is $L/M$-smooth.

When each $F_m$ is $\mu$-strongly convex,

$$
\langle \nabla H(X) - \nabla H(\hat{X}), X - \hat{X}\rangle = \left\langle \frac{1}{M}\begin{pmatrix} \nabla F_1(x_1)^\top - F_1(\hat{x}_1)^\top \\ \vdots \\ \nabla F_M(x_M)^\top - F_M(\hat{x}_M)^\top \end{pmatrix}, \begin{pmatrix} x_1^\top - \hat{x}_1^\top \\ \vdots \\ x_M^\top - \hat{x}_M^\top \end{pmatrix} \right\rangle
$$

$$
= \frac{1}{M}\sum_m [\nabla F_m(x_m) - F_m(\hat{x}_m)]^\top (x_m - \hat{x}_m)
$$

$$
\geq \frac{1}{M}\sum_m \mu\|x_m - \hat{x}_m\|^2 = \frac{\mu}{M}\|X - \hat{X}\|^2.
$$

Therefore, $H$ is $\mu/M$-strongly convex.

$\square$

**Proof for Proposition 2.3**

*Proof.* Here we use $\sigma_{\min}$ to denote smallest nonzero singular values of $A^\top$. We use Frobenius norm and the inner product $\langle A, B\rangle = Tr(A^\top B)$ by default.
**Smoothness:** Denote $x^*(\lambda) = \min_x f(x) + \langle \lambda, Ax\rangle$. By the optimality condition, for any $\lambda_1$ and $\lambda_2$, we have

$$
\langle x - x^*(\lambda_1), \nabla f(x^*(\lambda_1)) + A^\top\lambda_1\rangle \geq 0, \quad \forall x;
$$
$$
\langle x - x^*(\lambda_2), \nabla f(x^*(\lambda_2)) + A^\top\lambda_2\rangle \geq 0, \quad \forall x.
$$

Plugging in $x = x^*(\lambda_2)$ in the first inequality and $x = x^*(\lambda_1)$ in the second inequality, and adding them,

$$
\langle x^*(\lambda_2) - x^*(\lambda_1), \nabla f(x^*(\lambda_1)) - \nabla f(x^*(\lambda_2)) + A^\top\lambda_1 - A^\top\lambda_2\rangle \geq 0. \tag{6}
$$

Because $f$ is $\nu$-strongly convex,

$$
\langle x^*(\lambda_2) - x^*(\lambda_1), \nabla f(x^*(\lambda_2)) - \nabla f(x^*(\lambda_1))\rangle \geq \nu\|x^*(\lambda_2) - x^*(\lambda_1)\|^2.
$$

Combining the previous two inequalities,

$$
\langle x^*(\lambda_2) - x^*(\lambda_1), A^\top\lambda_1 - A^\top\lambda_2\rangle \geq \nu\|x^*(\lambda_2) - x^*(\lambda_1)\|^2.
$$

Noting that the right hand side

$$\langle x^*(\lambda_2) - x^*(\lambda_1), A^\top \lambda_1 - A^\top \lambda_2 \rangle \leq \|x^*(\lambda_2) - x^*(\lambda_1)\|\|A^\top(\lambda_1 - \lambda_2)\|$$
$$\leq \sigma_{\max}\|x^*(\lambda_2) - x^*(\lambda_1)\|\|\lambda_1 - \lambda_2\|.$$

Therefore,

$$\sigma_{\max}\|x^*(\lambda_2) - x^*(\lambda_1)\|\|\lambda_1 - \lambda_2\| \geq \nu\|x^*(\lambda_2) - x^*(\lambda_1)\|^2,$$

which implies

$$\|x^*(\lambda_2) - x^*(\lambda_1)\| \leq \frac{\sigma_{\max}}{\nu}\|\lambda_1 - \lambda_2\|.$$

This implies that $x^*(\lambda)$ is $\frac{\sigma_{\max}}{\lambda}$-Lipschitz. By Danskin's Theorem, $\nabla h(\lambda) = -Ax^*(\lambda)$. Then,

$$\|\nabla h(\lambda_1) - \nabla h(\lambda_2)\| = \|Ax^*(\lambda_1) - Ax^*(\lambda_2)\| \leq \sigma_{\max}\|x^*(\lambda_1) - x^*(\lambda_2)\| \leq \frac{\sigma_{\max}^2}{\nu}\|\lambda_1 - \lambda_2\|.$$

We conclude that $h$ is $\frac{\sigma_{\max}^2}{\nu}$-smooth.

**Strong concavity:** Since $f$ is $\ell$-smooth, for any $\lambda_1$ and $\lambda_2$,

$$\langle x^*(\lambda_2) - x^*(\lambda_1), \nabla f(x^*(\lambda_2)) - \nabla f(x^*(\lambda_1)) \rangle \geq \frac{1}{\ell}\|\nabla f(x^*(\lambda_2)) - \nabla f(x^*(\lambda_1))\|^2$$
$$= \frac{1}{\ell}\|A^\top\lambda_1 - A^\top\lambda_2\|^2,$$

where in the equality, we use $\nabla f(x^*(\lambda)) = -A^\top\lambda$ by the optimality of $x^*(\lambda)$. For $\lambda_1, \lambda_2 \in \text{span}(A)$, we have $\lambda_1 - \lambda_2 \in \text{span}(A)$. Let the compact SVD of $A^\top$ be $A^\top = U\Sigma V^\top$. Then

$$\|A^\top(\lambda_1 - \lambda_2)\|^2 = \sum_{i=1}^{n}(\lambda_1 - \lambda_2)_i^\top AA^\top(\lambda_1 - \lambda_2)_i$$
$$= \sum_{i=1}^{n}\left(V^\top(\lambda_1 - \lambda_2)_i\right)^\top \Sigma^2 \left(V^\top(\lambda_1 - \lambda_2)_i\right)$$
$$\geq \sigma_{\min}^2\|V^\top(\lambda_1 - \lambda_2)\|^2 = \sigma_{\min}^2\|\lambda_1 - \lambda_2\|^2,$$

where in the last equality we use $\lambda_1 - \lambda_2 \in \text{span}(A)$. Combining with the last inequality,

$$\langle x^*(\lambda_2) - x^*(\lambda_1), \nabla f(x^*(\lambda_2)) - \nabla f(x^*(\lambda_1)) \rangle \geq \frac{\sigma_{\min}^2}{\ell}\|\lambda_1 - \lambda_2\|^2.$$

Combining with (6),

$$\langle x^*(\lambda_2) - x^*(\lambda_1), A^\top\lambda_1 - A^\top\lambda_2 \rangle \geq \frac{\sigma_{\min}^2}{\ell}\|\lambda_1 - \lambda_2\|^2.$$

We further note that

$$\langle \nabla h(\lambda_2) - \nabla h(\lambda_1), \lambda_1 - \lambda_2 \rangle = \langle Ax^*(\lambda_2) - Ax^*(\lambda_1), \lambda_1 - \lambda_2 \rangle = \langle x^*(\lambda_2) - x^*(\lambda_1), A^\top\lambda_1 - A^\top\lambda_2 \rangle.$$

Therefore,

$$\langle \nabla h(\lambda_2) - \nabla h(\lambda_1), \lambda_1 - \lambda_2 \rangle \geq \frac{\sigma_{\min}^2}{\ell}\|\lambda_1 - \lambda_2\|^2.$$

We conclude that $h$ is $\frac{\sigma_{\min}^2}{\ell}$-strongly concave. $\qquad\square$

# B. Proofs for Inexact Accelerated Gradient Method and Reproducibility

We restate Theorem 3.2.

**Theorem B.1.** *Suppose that $f$ is $\mu$-strongly convex and $\ell$-smooth over a closed convex set $\mathcal{X}$, and that we have access to a $\delta$-inexact gradient oracle as in Definition 3.1. Starting from $y_0 = x_0$, and using a stepsize $\tau \leq \frac{1}{2\ell}$, Inexact-AGD with $\beta = 0$ satisfies, for any $\theta \in (0, \frac{\mu\tau}{2}]$,*

$$\Delta_{t+1} \leq (1-\theta)\Delta_t + \left(\tau + \frac{\theta}{\mu}\right)\delta^2,$$

*and for $\theta \in \left(\frac{\mu\tau}{2}, \sqrt{\frac{\mu\tau}{2}}\right]$, with $\beta = \frac{(2\theta-\mu\tau)(1-\theta)}{2\theta-\mu\tau\theta}$, it holds that*

$$\Delta_{t+1} + \frac{2\theta^2 - \theta\mu\tau}{4\tau(1-\theta)}\|u_{t+1} - x^*\|^2 \leq (1-\theta)\left(\Delta_t + \frac{2\theta^2 - \theta\mu\tau}{4\tau(1-\theta)}\|u_t - x^*\|^2\right) + \left(\frac{\tau}{2} + \frac{2\theta}{\mu}\right)\delta^2,$$

*where $\Delta_t = f(x_t) - \min_{x \in \mathcal{X}} f(x)$ and $u_t = \frac{1}{\theta}x_t + \frac{1-\theta}{\theta}x_{t-1}$.*

*Proof.* By smoothness and definition of $\delta$-inexact gradient, we have that

$$f(x_{t+1}) \leq f(y_t) + \nabla f(y_t)^\top (x_{t+1} - y_t) + \frac{\ell}{2}\|x_{t+1} - y_t\|^2$$

$$= f(y_t) + g(y_t)^\top (x_{t+1} - y_t) + (\nabla f(y_t) - g(y_t))^\top (x_{t+1} - y_t) + \frac{\ell}{2}\|x_{t+1} - y_t\|^2$$

$$\leq f(y_t) + g(y_t)^\top (x_{t+1} - y_t) + \tau\|\nabla f(y_t) - g(y_t)\|^2 + \frac{1}{4\tau}\|x_{t+1} - y_t\|^2 + \frac{\ell}{2}\|x_{t+1} - y_t\|^2$$

$$\leq f(y_t) + g(y_t)^\top (x_{t+1} - y_t) + \left(\frac{\ell}{2} + \frac{1}{4\tau}\right)\|x_{t+1} - y_t\|^2 + \tau\delta^2. \tag{7}$$

By strong convexity and inexact gradient,

$$f(x^*) \geq f(y_t) + \nabla f(y_t)^\top (x^* - y_t) + \frac{\mu}{2}\|x^* - y_t\|^2$$

$$= f(y_t) + g(y_t)^\top (x^* - y_t) + (\nabla f(y_t) - g(y_t))^\top (x^* - y_t) + \frac{\mu}{2}\|x^* - y_t\|^2$$

$$\geq f(y_t) + g(y_t)^\top (x^* - y_t) - \frac{1}{\mu}\|\nabla f(y_t) - g(y_t)\|^2 - \frac{\mu}{4}\|x^* - y_t\|^2 + \frac{\mu}{2}\|x^* - y_t\|^2$$

$$\geq f(y_t) + g(y_t)^\top (x^* - y_t) + \frac{\mu}{4}\|x^* - y_t\|^2 - \frac{\delta^2}{\mu} \tag{8}$$

Similarly,

$$f(x_t) \geq f(y_t) + g(y_t)^\top (x_t - y_t) + (\nabla f(y_t) - g(y_t))^\top (x_t - y_t) + \frac{\mu}{2}\|x_t - y_t\|^2 \tag{9}$$

Combing (7) and (8),

$$f(x_{t+1}) - f(x^*)$$
$$= f(x_{t+1}) - f(y_t) + f(y_t) - f(x^*)$$
$$\leq g(y_t)^\top (x_{t+1} - x^*) + \left(\frac{\ell}{2} + \frac{1}{4\tau}\right)\|x_{t+1} - y_t\|^2 - \frac{\mu}{4}\|x^* - y_t\|^2 + \left(\tau + \frac{1}{\mu}\right)\delta^2$$

$$\leq -\frac{1}{\tau}(x_{t+1} - y_t)^\top (x_{t+1} - x^*) + \left(\frac{\ell}{2} + \frac{1}{4\tau}\right)\|x_{t+1} - y_t\|^2 - \frac{\mu}{4}\|x^* - y_t\|^2 + \left(\tau + \frac{1}{\mu}\right)\delta^2$$

$$= \frac{1}{\tau}(x_{t+1} - y_t)^\top (x^* - y_t) - \frac{1}{\tau}\|x_{t+1} - y_t\|^2 + \left(\frac{\ell}{2} + \frac{1}{4\tau}\right)\|x_{t+1} - y_t\|^2 - \frac{\mu}{4}\|x^* - y_t\|^2 + \left(\tau + \frac{1}{\mu}\right)\delta^2 \tag{10}$$

$$\leq -\frac{1}{2\tau}\|x_{t+1} - y_t\|^2 + \frac{1}{\tau}(x_{t+1} - y_t)^\top (x^* - y_t) - \frac{\mu}{4}\|x^* - y_t\|^2 + \left(\tau + \frac{1}{\mu}\right)\delta^2, \tag{11}$$

where in the last inequality we use $\frac{1}{\tau} - \frac{\ell}{2} - \frac{1}{4\tau} \geq \frac{1}{2\tau}$ as $\tau \leq \frac{1}{2\ell}$; in the second to last inequality we use the optimality condition of the projection step that $\forall \, x \in \mathcal{X}$,

$$\left( x_{t+1} - y_t + \frac{1}{2\ell} g(y_t) \right)^\top (x - x_{t+1}) \geq 0.$$

Similarly, combining (7) and (9),

$$f(x_{t+1}) - f(x_t) \leq -\frac{1}{2\tau} \|x_{t+1} - y_t\|^2 + \frac{1}{\tau}(x_{t+1} - y_t)^\top (x_t - y_t) + \tau\delta^2$$
$$- (\nabla f(y_t) - g(y_t))^\top (x_t - y_t) - \frac{\mu}{2}\|x_t - y_t\|^2. \tag{12}$$

Let $\theta > 0$ and $\Delta_t := f(x_t) - f(x^*) \geq 0$. We will pick $\theta$ later. Compute $\theta*(11) +(1 - \theta)*(12)$, leading to

$$\Delta_{t+1} - (1 - \theta)\Delta_t$$
$$\leq -\frac{1}{2\tau}\|x_{t+1} - y_t\|^2 + \frac{1}{\tau}(x_{t+1} - y_t)^\top((1 - \theta)x_t + \theta x^* - y_t) - \frac{\mu}{4}\theta\|x^* - y_t\|^2$$
$$- \frac{\mu(1 - \theta)}{2}\|x_t - y_t\|^2 + \left(\tau + \frac{\theta}{\mu}\right)\delta^2 - (1 - \theta)(\nabla f(y_t) - g(y_t))^\top(x_t - y_t)$$
$$= \frac{1}{2\tau}\|y_t - (1 - \theta)x_t - \theta x^*\|^2 - \frac{1}{2\tau}\|x_{t+1} - (1 - \theta)x_t - \theta x^*\|^2 - \frac{\mu}{4}\theta\|x^* - y_t\|^2$$
$$- \frac{\mu(1 - \theta)}{2}\|x_t - y_t\|^2 + \left(\tau + \frac{\theta}{\mu}\right)\delta^2 - (1 - \theta)(\nabla f(y_t) - g(y_t))^\top(x_t - y_t) \tag{13}$$

Let $\theta u_t := x_t - (1 - \theta)x_{t-1}$. From the update of Inexact-AGD, we observe

$$y_t = x_t + \beta(x_t - x_{t-1})$$
$$= (1 + \beta)x_t - \beta x_{t-1}$$
$$= (1 + \beta)x_t - \beta\left(\frac{x_t}{1 - \theta} - \frac{\theta}{1 - \theta}u_t\right)$$
$$= \left(1 + \beta - \frac{\beta}{1 - \theta}\right)x_t + \frac{\beta\theta}{1 - \theta}u_t \tag{14}$$

Rearranging terms, we can get $x_t = \frac{1}{1+\beta-\frac{\beta}{1-\theta}}\left(y_t - \frac{\beta\theta}{1-\theta}u_t\right)$ and thus

$$y_t - (1 - \theta)x_t = y_t - (1 - \theta)\frac{1}{1 + \beta - \frac{\beta}{1-\theta}}\left(y_t - \frac{\beta\theta}{1 - \theta}u_t\right)$$
$$= \left(1 - \frac{1 - \theta}{1 + \beta - \frac{\beta}{1-\theta}}\right)y_t + \frac{\beta\theta}{1 + \beta - \frac{\beta}{1-\theta}}u_t$$
$$= \theta\left[\left(\frac{1}{\theta} - \frac{1 - \theta}{\theta\left(1 + \beta - \frac{\beta}{1-\theta}\right)}\right)y_t + \frac{\beta\theta}{1 + \beta - \frac{\beta}{1-\theta}}u_t\right]$$
$$= \theta\left[\left(1 - \frac{\beta}{1 + \beta - \frac{\beta}{1-\theta}}\right)y_t + \frac{\beta\theta}{1 + \beta - \frac{\beta}{1-\theta}}u_t\right], \tag{15}$$

where in the last equality we note that

$$
\begin{aligned}
\frac{1}{\theta} - \frac{1-\theta}{\theta\left(1+\beta-\frac{\beta}{1-\theta}\right)} &= \frac{1+\beta-\frac{\beta}{1-\theta}-1+\theta}{\theta\left(1+\beta-\frac{\beta}{1-\theta}\right)} = \frac{\beta-\frac{\beta}{1-\theta}+\theta}{\theta\left(1+\beta-\frac{\beta}{1-\theta}\right)} \\
&= 1 - \frac{\theta+\theta\beta-\frac{\theta\beta}{1-\theta}-\beta+\frac{\beta}{1-\theta}-\theta}{\theta\left(1+\beta-\frac{\beta}{1-\theta}\right)} = 1 - \frac{\theta\beta-\frac{\theta\beta}{1-\theta}-\beta+\frac{\beta}{1-\theta}}{\theta\left(1+\beta-\frac{\beta}{1-\theta}\right)} \\
&= 1 - \frac{(\theta-1)\beta-\frac{(\theta-1)\beta}{1-\theta}}{\theta\left(1+\beta-\frac{\beta}{1-\theta}\right)} = 1 - \frac{\beta}{1+\beta-\frac{\beta}{1-\theta}}
\end{aligned}
$$

It is easy to verify that the above also holds when $u_0 = x_0 = y_0$. Define $v = \frac{\beta}{1+\beta-\frac{\beta}{1-\theta}}$. Plug (15) into (13),

$$
\begin{aligned}
&\Delta_{t+1} - (1-\theta)\Delta_t \\
&\leq \frac{\theta^2}{2\tau}\|(1-v)y_t + vu_t - x^*\|^2 - \frac{\theta^2}{2\tau}\|u_{t+1} - x^*\|^2 - \frac{\mu}{4}\theta\|x^* - y_t\|^2 - \\
&\quad \frac{\mu(1-\theta)}{2}\|x_t - y_t\|^2 + \left(\tau+\frac{\theta}{\mu}\right)\delta^2 - (1-\theta)(\nabla f(y_t) - g(y_t))^\top(x_t - y_t) \\
&= \frac{\theta^2}{2\tau}\|(1-v)(y_t - x^*) + v(u_t - x^*)\|^2 - \frac{\theta^2}{2\tau}\|u_{t+1} - x^*\|^2 - \frac{\mu}{4}\theta\|x^* - y_t\|^2 - \\
&\quad \frac{\mu(1-\theta)}{2}\|x_t - y_t\|^2 + \left(\tau+\frac{\theta}{\mu}\right)\delta^2 - (1-\theta)(\nabla f(y_t) - g(y_t))^\top(x_t - y_t) \\
&= \frac{\theta^2}{2\tau}\left[(1-v)^2\|y_t - x^*\|^2 + v^2\|u_t - x^*\|^2 + 2v(1-v)(y_t - x^*)^\top(u_t - x^*)\right] - \\
&\quad \frac{\theta^2}{2\tau}\|u_{t+1} - x^*\|^2 - \frac{\mu}{4}\theta\|x^* - y_t\|^2 - \frac{\mu(1-\theta)}{2}\|x_t - y_t\|^2 + \\
&\quad \left(\tau+\frac{\theta}{\mu}\right)\delta^2 - (1-\theta)(\nabla f(y_t) - g(y_t))^\top(x_t - y_t) \\
&= \frac{\theta^2}{2\tau}\left[(1-v)^2\|y_t - x^*\|^2 + v^2\|u_t - x^*\|^2 + v(1-v)\left(\|y_t - x^*\|^2 + \|u_t - x^*\|^2 - \|y_t - u_t\|^2\right)\right] - \\
&\quad \frac{\theta^2}{2\tau}\|u_{t+1} - x^*\|^2 - \frac{\mu}{4}\theta\|x^* - y_t\|^2 - \frac{\mu(1-\theta)}{2}\|x_t - y_t\|^2 + \\
&\quad \left(\tau+\frac{\theta}{\mu}\right)\delta^2 - (1-\theta)(\nabla f(y_t) - g(y_t))^\top(x_t - y_t) \\
&= \frac{\theta^2}{2\tau}\left[(1-v)^2 + v(1-v)\right]\|y_t - x^*\|^2 + \frac{\theta^2}{2\tau}\left[v^2 + v(1-v)\right]\|u_t - x^*\|^2 - \frac{\theta^2}{2\tau}v(1-v)\|y_t - u_t\|^2 - \\
&\quad \frac{\theta^2}{2\tau}\|u_{t+1} - x^*\|^2 - \frac{\mu}{4}\theta\|x^* - y_t\|^2 - \frac{\mu(1-\theta)}{2}\|x_t - y_t\|^2 + \\
&\quad \left(\tau+\frac{\theta}{\mu}\right)\delta^2 - (1-\theta)(\nabla f(y_t) - g(y_t))^\top(x_t - y_t)
\end{aligned}
$$

where in the second equality we use $\|(1-v)a + vb\|^2 = (1-v)^2\|a\|^2 + v^2\|b\|^2 + 2v(1-v)a^\top b$, and in the third equality we use $2a^\top b = \|a\|^2 + \|b\|^2 - \|a-b\|^2$. Rearranging (14)

$$
y_t = \frac{1-\theta-\beta\theta}{1-\theta}x_t + \frac{\beta\theta}{1-\theta}u_t \iff u_t - y_t = \frac{1-\theta-\beta\theta}{\beta\theta}(y_t - x_t)
$$

Plugging into the previous inequality, we then obtain

$$
\Delta_{t+1} + \frac{\theta^2}{2\tau}\|u_{t+1} - x^*\|^2 \le (1-\theta)\left(\Delta_t + \frac{\theta^2\left[v^2 + v(1-v)\right]}{2\tau(1-\theta)}\|u_t - x^*\|^2\right) + \left(\tau + \frac{\theta}{\mu}\right)\delta^2 -
$$
$$
\left[\frac{\mu\theta}{4} - \frac{\theta^2}{2\tau}\left[(1-v)^2 + v(1-v)\right]\right]\|y_t - x^*\|^2 -
$$
$$
\left[\frac{\theta^2}{2\tau}v(1-v)\left(\frac{1-\theta}{\beta\theta} - 1\right)^2 + \frac{\mu(1-\theta)}{2}\right]\|y_t - x_t\|^2 -
$$
$$
(1-\theta)(\nabla f(y_t) - g(y_t))^\top(x_t - y_t). \tag{16}
$$

To have a contraction, we want to choose $v$ such that the following holds,

$$
\frac{\mu\theta}{4} - \frac{\theta^2}{2\tau}\left[(1-v)^2 + v(1-v)\right] \ge 0 \tag{A1}
$$

$$
\frac{\theta^2\left[v^2 + v(1-v)\right]}{2\tau(1-\theta)} \le \frac{\theta^2}{2\tau}. \tag{B1}
$$

From (B1), we have

$$
(B1) \Longleftrightarrow v^2 + v(1-v) \le 1 - \theta \Longleftrightarrow v \le 1 - \theta \Longleftrightarrow \beta \le \frac{1-\theta}{1+\theta}. \tag{17}
$$

From (A1), we have

$$
(A1) \Longleftrightarrow \frac{\mu\theta}{4} \ge \frac{\theta^2}{2\tau}\left[(1-v)^2 + v(1-v)\right] \Longleftrightarrow 1 - v \le \frac{\mu\tau}{2\theta} \Longleftrightarrow v \ge 1 - \frac{\mu\tau}{2\theta}. \tag{18}
$$

From the upper and lower bounds for $v$ in (17) and (18), we require

$$
1 - \frac{\mu\tau}{2\theta} \le 1 - \theta \text{ and } 0 \le \theta \le 1 \Longleftrightarrow 0 \le \theta \le \sqrt{\frac{\mu\tau}{2}}. \tag{19}
$$

We further pick $\beta$ from (18):

$$
\text{If } \theta \le \frac{\mu\tau}{2}, \text{ then we choose } v = 0 \Longleftrightarrow \beta = 0; \tag{20}
$$

$$
\text{If } \theta > \frac{\mu\tau}{2}, \text{ then we choose } v = 1 - \frac{\mu\tau}{2\theta} \Longleftrightarrow \frac{\beta}{1 + \beta - \frac{\beta}{1-\theta}} = 1 - \frac{\mu\tau}{2\theta}
$$

$$
\Longleftrightarrow \beta = 1 + \beta - \frac{\beta}{1-\theta} - \frac{\mu\tau}{2\theta} - \frac{\beta\mu\tau}{2\theta} + \frac{\beta\mu\tau}{2\theta(1-\theta)}
$$

$$
\Longleftrightarrow 1 - \frac{\beta}{1-\theta} - \frac{\mu\tau}{2\theta} - \frac{\beta\mu\tau}{2\theta} + \frac{\beta\mu\tau}{2\theta(1-\theta)} = 0
$$

$$
\Longleftrightarrow \beta\left(\frac{1}{1-\theta} + \frac{\mu\tau}{2\theta} - \frac{\mu\tau}{2\theta(1-\theta)}\right) = 1 - \frac{\mu\tau}{2\theta} \Longleftrightarrow \beta \cdot \frac{2\theta - \mu\tau\theta}{2\theta(1-\theta)} = \frac{2\theta - \mu\tau}{2\theta}
$$

$$
\Longleftrightarrow \beta = \frac{2\theta - \mu\tau}{2\theta - \mu\tau\theta}(1-\theta). \tag{21}
$$

In the first case (20), when we choose $v = \beta = 0$, we have $x_t = y_t$ and (16) becomes

$$
\Delta_{t+1} + \frac{\theta^2}{2\tau}\|u_{t+1} - x^*\|^2 \le (1-\theta)\Delta_t + \left(\tau + \frac{\theta}{\mu}\right)\delta^2 - \left[\frac{\mu\theta}{4} - \frac{\theta^2}{2\tau}\right]\|y_t - x^*\|^2
$$

$$
\Longrightarrow \Delta_{t+1} \le (1-\theta)\Delta_t + \left(\tau + \frac{\theta}{\mu}\right)\delta^2. \tag{22}
$$

In the second case (21), by our choice of $v$ and $\beta$, we note that the coefficient for the second negative term in (16) becomes

$$
\frac{\theta^2}{2\tau}v(1-v)\left(\frac{1-\theta}{\beta\theta}-1\right)^2 + \frac{\mu(1-\theta)}{2}
$$
$$
= \frac{\theta^2}{2\tau}\left(1-\frac{\mu\tau}{2\theta}\right)\frac{\mu\tau}{2\theta}\left(\frac{2-\mu\tau}{2\theta-\mu\tau}-1\right)^2 + \frac{\mu(1-\theta)}{2}
$$
$$
= \frac{\theta^2}{2\tau}\cdot\frac{2\theta-\mu\tau}{2\theta}\cdot\frac{\mu\tau}{2\theta}\cdot\frac{[2(1-\theta)]^2}{(2\theta-\mu\tau)^2} + \frac{\mu(1-\theta)}{2}
$$
$$
= \frac{\mu(1-\theta)^2}{2(2\theta-\mu\tau)} + \frac{\mu(1-\theta)}{2}. \tag{23}
$$

The last two terms in (16) becomes

$$
-\left[\frac{\theta^2}{2\tau}v(1-v)\left(\frac{1-\theta}{\beta\theta}-1\right)^2 + \frac{\mu(1-\theta)}{2}\right]\|y_t-x_t\|^2 - (1-\theta)(\nabla f(y_t)-g(y_t))^\top(x_t-y_t)
$$
$$
= -\left[\frac{\mu(1-\theta)^2}{2(2\theta-\mu\tau)} + \frac{\mu(1-\theta)}{2}\right]\|y_t-x_t\|^2 - (1-\theta)(\nabla f(y_t)-g(y_t))^\top(x_t-y_t)
$$
$$
\leq \frac{2\theta-\mu\tau}{2\mu}\|\nabla f(y_t)-g(y_t)\|^2 \leq \frac{2\theta-\mu\tau}{2\mu}\delta^2. \tag{24}
$$

Plugging back into (16), as $v = 1-\frac{\mu\tau}{2\theta}$,

$$
\Delta_{t+1} + \frac{\theta^2}{2\tau}\|u_{t+1}-x^*\|^2 \leq (1-\theta)\left(\Delta_t + \frac{\ell\theta^2\left(1-\frac{\mu\tau}{2\theta}\right)}{1-\theta}\|u_t-x^*\|^2\right) + \left(\tau + \frac{\theta}{\mu} + \frac{2\theta-\mu\tau}{2\mu}\right)\delta^2
$$
$$
\implies \Delta_{t+1} + \frac{2\theta^2-\theta\mu\tau}{4\tau(1-\theta)}\|u_{t+1}-x^*\|^2 \leq (1-\theta)\left(\Delta_t + \frac{2\theta^2-\theta\mu\tau}{4\tau(1-\theta)}\|u_t-x^*\|^2\right) + \left(\frac{\tau}{2} + \frac{2\theta}{\mu}\right)\delta^2
$$

$\square$

## B.1. Proofs for Reproducibility

We restate Theorem 3.5.

**Theorem B.2.** *Assume $f$ is convex and $\ell$-smooth. Let $\|x_0 - x^*\| \leq D$ for some optimal solution $x^*$, and suppose $0 < \epsilon \leq \ell D^2$. Given a $\delta$-inexact gradient oracle, Algorithm 2 with $r = \epsilon/D^2$ and $\epsilon_r = 3\delta^2/r$ returns a point $x_r$ that is $(3\delta^2 D^2/\epsilon + \epsilon/2)$-optimal, with gradient complexity $\tilde{\mathcal{O}}\left(\sqrt{\ell D^2/\epsilon}\right)$. Moreover, the reproducibility is $\mathcal{O}(\delta^2/\epsilon^2)$.*

*Proof.* For $x^* \in \arg\min_{x\in\mathcal{X}} F(x)$ and $x_r^* = \arg\min_{x\in\mathcal{X}} F_r(x)$, we have that

$$
F(x_r) - F(x^*) \leq F_r(x_r) - F_r(x_r^*) + \frac{rD^2}{2}.
$$

For the reproducibility guarantee, using $r$-strong-convexity of $F_r(x)$, we can obtain that

$$
\|x_r - x_r'\| \leq \|x_r - x_r^*\| + \|x_r^* - x_r'\|
$$
$$
\leq \sqrt{\frac{2(F_r(x_r)-F_r(x_r^*))}{r}} + \sqrt{\frac{2(F_r(x_r')-F_r(x_r^*))}{r}}.
$$

Applying Corollary 3.3, since $r = \epsilon/D^2 \leq \ell$, we know that

$$
F_r(x_r) - F_r(x_r^*) \leq \left(1-\sqrt{\frac{r}{\ell+r}}\right)^T\left(F_r(x_0)-F_r(x_r^*)+\frac{r}{4}\|x_0-x_r^*\|^2\right) + \left(\frac{1}{2\sqrt{(\ell+r)r}}+\frac{1}{r}\right)\delta^2
$$
$$
\leq \left(1-\sqrt{\frac{r}{\ell+r}}\right)^T\left(F_r(x_0)-F_r(x_r^*)+\frac{r}{4}\|x_0-x_r^*\|^2\right) + \frac{2\delta^2}{r}.
$$

---

**Algorithm 3** Centralized Accelerated Gradient Ascent Multi-Stochastic Gradient Descent

---

1: **Input:** $x_0, \{\zeta_0^m\}_{m=1}^M$, stepsizes $\tau_1, \tau_2$, momentum $\beta$, inner steps $K$, outer steps $T$
2: **Initialize:** $x_{0,0}^m = x_0$ **and** $\tilde{\zeta}_0^m = \zeta_0^m$ **for all** $m \in [M]$
3: **for** $t = 0, 1, \ldots, T - 1$ **do**
4:     **for** $k = 0, 1, \ldots, K - 1$ **do**
5:         **for all** $m \in [M]$ **do**
6:             sample $\xi_k^m$
7:             $x_{t,k+1}^m = x_{t,k}^m - \tau_1 \left[ \frac{1}{M} g_m(x_{t,k}^m; \xi_k^m) + \tilde{\zeta}_t^m \right]$
8:         **end for**
9:     **end for**
10:    **for all** $m \in [M]$ **do**
11:       $x_{t+1,0}^m = x_{t,K}^m$
12:    **end for**
13:    $\bar{x}_{t+1,0} = \frac{1}{M} \sum_{j=1}^M x_{t+1,0}^j$
14:    **for all** $m \in [M]$ **do**
15:       $\zeta_{t+1}^m = \tilde{\zeta}_t^m + \tau_2 \left( x_{t+1,0}^m - \bar{x}_{t+1,0} \right)$
16:       $\tilde{\zeta}_{t+1}^m = \zeta_{t+1}^m + \beta \left( \zeta_{t+1}^m - \zeta_t^m \right)$
17:    **end for**
18: **end for**
19: **Output:** $\bar{x}_{T,K} = \frac{1}{M} \sum_{m=1}^M x_{T,K}^m$

---

When setting $T = \mathcal{O}(\sqrt{\ell/r} \log(r/\delta^2))$, the algorithm can converge to $F_r(x_r) - F_r(x_r^*) \leq 3\delta^2/r$ and therefore $\|x_r - x_r^*\|^2 \leq 6\delta^2/r^2$. Since we choose $r = \epsilon/D^2$,

$$F(x_r) - F(x^*) \leq \mathcal{O}\left( \frac{\delta^2}{\epsilon^2} + \epsilon \right),$$

and the reproducibility is $\|x_r - x_r'\|^2 \leq \mathcal{O}(\delta^2 D^2/\epsilon^2)$. $\qquad\square$

## C. Proofs for Local ADA

By Proposition A.1, the function $H(X) \triangleq \frac{1}{M} \sum_{m=1}^M F_m(x_m)$ is $(L/M)$-smooth and $(\mu/M)$-strongly convex with respect to $X$. For convenience, we denote $\mu_H \triangleq \mu/M$ and $L_H \triangleq L/M$. Moreover, the stochastic gradient $\widetilde{\nabla} H(X) \triangleq \frac{1}{M} \left( g_1(x_1; \xi_1), \ldots, g_M(x_M; \xi_M) \right)^\top$ is an unbiased estimator of $\nabla H(X)$ with variance $\sigma_H^2 \triangleq \sigma^2/M$. By Proposition 2.3, the dual function $\Psi(\lambda) \triangleq \min_X \mathcal{L}(X, \lambda)$ is $L_\Psi$-smooth and $\mu_\Psi$-strongly concave over $\mathrm{span}(U)$, where $L_\Psi = 2M/\mu$ and $\mu_\Psi = M(1 - \sigma_2)/L$.

We begin with an intermediate lemma establishing the contraction of an appropriately defined potential function, where the contraction rate depends on the algorithm parameters.

**Lemma C.1.** *Assume Assumptions 2.1 and 2.2 hold. We run Algorithm 1 with stepsizes $\tau_1$ and $\tau_2$, inner-loop steps $K$, and momentum parameter $\beta$ which is decided by $\theta > 0$. Denote $c = (1 - \mu_H \tau_2)^K$. If $\tau_1, \tau_2, p > 0$, $c$ and $\theta$ satisfy the following*

$$\tau_1 \leq \frac{1}{4L_\Psi}, \quad \tau_2 \leq \frac{1}{L_H}, \quad \sqrt{\frac{\mu_\Psi \tau_1}{2}} \geq \theta > \frac{\mu_\Psi \tau_1}{2}, \quad pc \leq \frac{1}{6}, \quad 2\theta - \mu_\Psi \tau_1 \leq \frac{\tau_1 \mu_\Psi}{32pc}, \quad \theta < \frac{\mu_H \mu_\Psi p}{28},$$

*then we have*

$$\mathbb{E}\left[ A_{t+1} + p\left( 1 - \frac{28\theta}{\mu_H \mu_\Psi p} \right) B_t + \frac{5pc}{4\tau_1} \|\lambda_{t+1} - \tilde{\lambda}_t\|^2 \right]$$

$$\leq \max\left\{ 1 - \theta, \frac{c\left( 1 + \frac{4\tau_1}{\mu_H} \right)}{1 - \frac{28\theta}{\mu_H \mu_\Psi p}}, \frac{1}{2} \right\} \mathbb{E}\left[ A_t + p\left( 1 - \frac{28\theta}{\mu_H \mu_\Psi p} \right) B_{t-1} + \frac{5pc}{4\tau_1} \mathbb{E}\|\tilde{\lambda}_{t-1} - \lambda_t\|^2 \right] + \frac{pL_H \tau_2^2 \sigma_H^2}{2} \sum_{k=0}^{K-1} (1 - \tau_2 \mu_H)^k,$$

*where $A_t \triangleq \Delta_t + \frac{\theta^2 v}{2\tau_1(1-\theta)}\|u_t - \lambda^*\|^2$, $\Delta_t \triangleq \Psi^* - \Psi(\lambda_{t+1})$, and $B_t \triangleq \mathbb{E}\left[\mathcal{L}(X_{t+1}, \tilde{\lambda}_t) - \Psi(\tilde{\lambda}_t)\right]$. Here, $u_t$ denotes a point on the line segment between $\lambda_t$ and $\lambda_{t-1}$, and $v$ is a scalar parameter; both are defined in (26).*

*Proof.* We decompose the proof into several parts. Algorithm 1 can be viewed as applying an accelerated method to maximize $\Psi$, using $\nabla_\lambda \mathcal{L}(X_{t+1}, \tilde{\lambda}_t)$ as an inexact gradient oracle. Consequently, several arguments follow directly from the proof of Theorem 3.2, with only minor modifications.

**Part I:**   Bounding the descent of accelerated gradient ascent (AGA).

We denote $\Delta_t = \Psi^* - \Psi(\lambda_{t+1})$, and $\delta_t = \|\nabla\Psi(\tilde{\lambda}_t) - \nabla_\lambda\mathcal{L}(X_{t+1}, \tilde{\lambda}_t)\|$. The analysis follows that of Theorem 3.2 up to equation (10). Using the stepsize condition $\tau_1 \leq 1/(4L_\Psi)$, we observe that $1/\tau_1 - L_\Psi/2 - 1/(4\tau_1) \geq 3/(4\tau_1)$. Analogously to (13), we obtain

$$
\begin{aligned}
&\Delta_{t+1} - (1-\theta)\Delta_t \\
&\leq -\frac{1}{2\tau_1}\|\lambda_{t+1} - \tilde{\lambda}_t\|^2 - \frac{1}{4\tau_1}\|\lambda_{t+1} - \tilde{\lambda}_t\|^2 + \frac{1}{\tau_1}(\lambda_{t+1} - \tilde{\lambda}_t)^\top((1-\theta)\lambda_t + \theta\lambda^* - \tilde{\lambda}_t) - \frac{\mu_\Psi\theta}{4}\|\lambda^* - \tilde{\lambda}_t\|^2 \\
&\quad - \frac{\mu_\Psi(1-\theta)}{2}\|\lambda_t - \tilde{\lambda}_t\|^2 + \left(\tau_1 + \frac{\theta}{\mu}\right)\delta_t^2 - (1-\theta)(\nabla_\lambda\mathcal{L}(X_{t+1}, \tilde{\lambda}_t) - \nabla\Psi(\tilde{\lambda}_t))^\top(\lambda_t - \tilde{\lambda}_t) \\
&= \frac{1}{2\tau_1}\|\tilde{\lambda}_t - (1-\theta)\lambda_t - \theta\lambda^*\|^2 - \frac{1}{2\tau_1}\|\lambda_{t+1} - (1-\theta)\lambda_t - \theta\lambda^*\|^2 - \frac{\mu_\Psi\theta}{4}\|\lambda^* - \tilde{\lambda}_t\|^2 - \frac{1}{4\tau_1}\|\lambda_{t+1} - \tilde{\lambda}_t\|^2 \\
&\quad - \frac{\mu_\Psi(1-\theta)}{2}\|\lambda_t - \tilde{\lambda}_t\|^2 + \left(\tau_1 + \frac{\theta}{\mu_\Psi}\right)\delta_t^2 - (1-\theta)(\nabla_\lambda\mathcal{L}(X_{t+1}, \tilde{\lambda}_t) - \nabla\Psi(\tilde{\lambda}_t))^\top(\lambda_t - \tilde{\lambda}_t) \quad (25)
\end{aligned}
$$

Define

$$
\theta u_t := \lambda_t - (1-\theta)\lambda_{t-1}, \quad v = \frac{\beta}{1 + \beta - \frac{\beta}{1-\theta}} \quad (26)
$$

Similar to (16),

$$
\begin{aligned}
\Delta_{t+1} + \frac{\theta^2}{2\tau_1}\|u_{t+1} - \lambda^*\|^2 \leq &\ (1-\theta)\left(\Delta_t + \frac{\theta^2 v}{2\tau_1(1-\theta)}\|u_t - \lambda^*\|^2\right) + \left(\tau_1 + \frac{\theta}{\mu_\Psi}\right)\delta_t^2 \\
&- \left[\frac{\mu_\Psi\theta}{4} - \frac{\theta^2}{2\tau_1}(1-v))\right]\|\tilde{\lambda}_t - \lambda^*\|^2 - \frac{1}{4\tau_1}\|\lambda_{t+1} - \tilde{\lambda}_t\|^2 \\
&- \left[\frac{\theta^2}{2\tau_1}v(1-v)\left(\frac{1-\theta}{\beta\theta} - 1\right)^2 + \frac{\mu_\Psi(1-\theta)}{2}\right]\|\tilde{\lambda}_t - \lambda_t\|^2 \\
&- (1-\theta)(\nabla_\lambda\mathcal{L}(X_{t+1}, \tilde{\lambda}_t) - \nabla\Psi(\tilde{\lambda}_t))^\top(\lambda_t - \tilde{\lambda}_t).
\end{aligned}
$$

**Part II:**   Bounding $\mathbb{E}[\mathcal{L}(X_{t+1}, \tilde{\lambda}_t) - \Psi(\tilde{\lambda}_t)]$.

Recall that $\delta_t = \|\nabla\Psi(\tilde{\lambda}_t) - \nabla_\lambda\mathcal{L}(X_{t+1}, \tilde{\lambda}_t)\|$. For convenience, we define $B_t = \mathbb{E}[\mathcal{L}(X_{t+1}, \tilde{\lambda}_t) - \Psi(\tilde{\lambda}_t)]$. The function $\mathcal{L}$ is $L_H$-smooth and $\mu_H$-strongly concave, where $L_H = \frac{L}{M}$ and $\mu_H = \frac{\mu}{M}$. By the result of stochastic gradient descent, after $K$ iterations of inner loops, with stepsize $\tau_2 \leq 1/L_H$,

$$
\begin{aligned}
B_t &= (1 - \mu_H\tau_2)^K \mathbb{E}\left[\mathcal{L}(X_t, \tilde{\lambda}_t) - \Psi(\tilde{\lambda}_t)\right] + \frac{L_H\tau_2^2\sigma_H^2}{2}\sum_{k=0}^{K-1}(1 - \tau_2\mu_H)^k \\
&= (1 - \mu_H\tau_2)^K \mathbb{E}\Big[\underbrace{\mathcal{L}(X_t, \tilde{\lambda}_{t-1}) - \Psi(\tilde{\lambda}_{t-1})}_{B_{t-1}} + \underbrace{\Psi(\tilde{\lambda}_{t-1}) - \Psi(\tilde{\lambda}_t)}_{C} \\
&\quad + \underbrace{\mathcal{L}(X_t, \tilde{\lambda}_t) - \mathcal{L}(X_t, \tilde{\lambda}_{t-1})}_{D}\Big] + \frac{L_H\tau_2^2\sigma_H^2}{2}\sum_{k=0}^{K-1}(1 - \tau_2\mu_H)^k \quad (27)
\end{aligned}
$$

We proceed to bound $C$ and $D$. Bound $D$ first,

$$\begin{aligned}
D &\triangleq \mathcal{L}(X_t, \tilde{\lambda}_t) - \mathcal{L}(X_t, \tilde{\lambda}_{t-1}) = \langle \tilde{\lambda}_t, UX_t \rangle - \langle \tilde{\lambda}_{t-1}, UX_t \rangle \\
&= \langle \lambda_t - \tilde{\lambda}_{t-1}, UX_t \rangle + \langle \tilde{\lambda}_t - \lambda_t, UX_t \rangle \\
&= \tau_1 \|UX_t\|^2 + \langle \tilde{\lambda}_t - \lambda_t, UX_t \rangle \\
&\leq \tau_1 \|UX_t\|^2 + \frac{1}{2a}\|\tilde{\lambda}_t - \lambda_t\|^2 + \frac{a}{2}\|UX_t\|^2 = \left(\tau_1 + \frac{a}{2}\right)\|UX_t\|^2 + \frac{1}{2a}\|\tilde{\lambda}_t - \lambda_t\|^2,
\end{aligned}$$
(28)

where in the last inequality, we use Young's inequality with $a > 0$. Then we bound $C$,

$$C \triangleq \Psi(\tilde{\lambda}_{t-1}) - \Psi(\tilde{\lambda}_t) = \underbrace{\Psi(\tilde{\lambda}_{t-1}) - \Psi(\lambda_t)}_{C_1} + \underbrace{\Psi(\lambda_t) - \Psi(\tilde{\lambda}_t)}_{C_2}.$$
(29)

We bound $C_1$ and $C_2$ separately. Similar to (7) (i.e., consider $-\Psi$ as $f$),

$$\begin{aligned}
C_1 &\triangleq \Psi(\tilde{\lambda}_{t-1}) - \Psi(\lambda_t) \\
&\leq \nabla_\lambda \mathcal{L}(X_t, \tilde{\lambda}_{t-1})^\top (\tilde{\lambda}_{t-1} - \lambda_t) + \left(\frac{L_\Psi}{2} + \frac{1}{4\tau_1}\right)\|\tilde{\lambda}_{t-1} - \lambda_t\|^2 + \tau_1 \delta_{t-1}^2 \\
&= \left(-\frac{1}{\tau_1} + \frac{L_\Psi}{2} + \frac{1}{4\tau_1}\right)\|\tilde{\lambda}_{t-1} - \lambda_t\|^2 + \tau_1 \delta_{t-1}^2 = \left(-\frac{3}{4\tau_1} + \frac{L_\Psi}{2}\right)\|\tilde{\lambda}_{t-1} - \lambda_t\|^2 + \tau_1 \delta_{t-1}^2,
\end{aligned}$$
(30)

where in the second equality we use $\lambda^t = \tilde{\lambda}^{t-1} + \tau_1 \nabla_\lambda \mathcal{L}(X_t, \tilde{\lambda}_{t-1})$. By strong concavity of $\Psi$,

$$\begin{aligned}
C_2 &\triangleq \Psi(\lambda_t) - \Psi(\tilde{\lambda}_t) \\
&\leq \nabla\Psi(\tilde{\lambda}_t)^\top (\lambda_t - \tilde{\lambda}_t) - \frac{\mu_\Psi}{2}\|\lambda_t - \tilde{\lambda}_t\|^2 \\
&= \nabla_\lambda \mathcal{L}(X_{t+1}, \tilde{\lambda}_{t+1})^\top (\lambda_t - \tilde{\lambda}_t) + \left(\nabla\Psi(\tilde{\lambda}_t) - \nabla_\lambda \mathcal{L}(X_{t+1}, \tilde{\lambda}_{t+1})\right)^\top (\lambda_t - \tilde{\lambda}_t) - \frac{\mu_\Psi}{2}\|\lambda_t - \tilde{\lambda}_t\|^2 \\
&\leq \frac{b}{2}\|\nabla_\lambda \mathcal{L}(X_{t+1}, \tilde{\lambda}_{t+1})\|^2 + \left(\frac{1}{2b} - \frac{\mu_\Psi}{2}\right)\|\lambda_t - \tilde{\lambda}_t\|^2 + \left(\nabla\Psi(\tilde{\lambda}_t) - \nabla_\lambda \mathcal{L}(X_{t+1}, \tilde{\lambda}_{t+1})\right)^\top (\lambda_t - \tilde{\lambda}_t),
\end{aligned}$$
(31)

where we use Young's inequality with $b > 0$ in the last inequality. Combining (27), (28), (29), (30) and (31),

$$\begin{aligned}
B_t &\leq (1 - \mu_H \tau_2)^K \mathbb{E}\Bigg[ B_{t-1} + \left(\tau_1 + \frac{a}{2}\right)\|UX_t\|^2 + \frac{1}{2a}\|\tilde{\lambda}_t - \lambda_t\|^2 + \left(-\frac{3}{4\tau_1} + \frac{L_\Psi}{2}\right)\|\tilde{\lambda}_{t-1} - \lambda_t\|^2 \\
&\quad + \tau_1 \delta_{t-1}^2 + \frac{b}{2}\|\nabla_\lambda \mathcal{L}(X_{t+1}, \tilde{\lambda}_{t+1})\|^2 + \left(\frac{1}{2b} - \frac{\mu_\Psi}{2}\right)\|\lambda_t - \tilde{\lambda}_t\|^2 \\
&\quad + \left(\nabla\Psi(\tilde{\lambda}_t) - \nabla_\lambda \mathcal{L}(X_{t+1}, \tilde{\lambda}_{t+1})\right)^\top (\lambda_t - \tilde{\lambda}_t) \Bigg] + \frac{L_H \tau_2^2 \sigma_H^2}{2} \sum_{k=0}^{K-1} (1 - \tau_2 \mu_H)^k.
\end{aligned}$$

Note that

$$\delta_{t-1}^2 = \|\nabla\Psi(\tilde{\lambda}_{t-1}) - \nabla_\lambda \mathcal{L}(X_t, \tilde{\lambda}_{t-1})\|^2 = \|UX_t - UX^*(\tilde{\lambda}_{t-1})\|^2 \leq 2\|X^*(\tilde{\lambda}_{t-1}) - X_t\|^2 \leq \frac{4}{\mu_H} B_{t-1};$$

$$\|\nabla_\lambda \mathcal{L}(X_{t+1}, \tilde{\lambda}_{t+1})\| = \|UX_{t+1}\|^2 = \frac{1}{\tau_1^2}\|\lambda_{t+1} - \tilde{\lambda}_t\|^2.$$

Plugging back into the previous inequality,

$$
\begin{aligned}
B_t \leq{} & (1 - \mu_H \tau_2)^K \mathbb{E}\Bigg[ \left(1 + \frac{4\tau_1}{\mu_H}\right) B_{t-1} + \left(\frac{1}{\tau_1} + \frac{a}{2\tau_1^2} - \frac{3}{4\tau_1} + \frac{L_\Psi}{2}\right) \|\tilde{\lambda}_{t-1} - \lambda_t\|^2 \\
& + \frac{b}{2\tau_1^2} \|\lambda_{t+1} - \tilde{\lambda}_t\|^2 + \left(\frac{1}{2a} + \frac{1}{2b} - \frac{\mu_\Psi}{2}\right) \|\lambda_t - \tilde{\lambda}_t\|^2 \\
& + \left(\nabla\Psi(\tilde{\lambda}_t) - \nabla_\lambda \mathcal{L}(X_{t+1}, \tilde{\lambda}_{t+1})\right)^\top (\lambda_t - \tilde{\lambda}_t)\Bigg] + \frac{L_H \tau_2^2 \sigma_H^2}{2} \sum_{k=0}^{K-1} (1 - \tau_2 \mu_H)^k \\
={} & (1 - \mu_H \tau_2)^K \mathbb{E}\Bigg[ \left(1 + \frac{4\tau_1}{\mu_H}\right) B_{t-1} + \left(\frac{1}{4\tau_1} + \frac{a}{2\tau_1^2} + \frac{L_\Psi}{2}\right) \|\tilde{\lambda}_{t-1} - \lambda_t\|^2 + \frac{a}{2\tau_1^2} \|\lambda_{t+1} - \tilde{\lambda}_t\|^2 + \\
& \left(\frac{1}{a} - \frac{\mu_\Psi}{2}\right) \|\lambda_t - \tilde{\lambda}_t\|^2 + \left(\nabla\Psi(\tilde{\lambda}_t) - \nabla_\lambda \mathcal{L}(X_{t+1}, \tilde{\lambda}_{t+1})\right)^\top (\lambda_t - \tilde{\lambda}_t)\Bigg] + \frac{L_H \tau_2^2 \sigma_H^2}{2} \sum_{k=0}^{K-1} (1 - \tau_2 \mu_H)^k,
\end{aligned}
\tag{32}
$$

where in the equality we choose $a = b$.

**Part III:** Establishing potential functions.

Combining Part I and II, for $p > 0$,

$$
\begin{aligned}
& \mathbb{E}\left[ \Delta_{t+1} + \frac{\theta^2}{2\tau_1} \|u_{t+1} - \lambda^*\|^2 + p\left(\mathcal{L}(X_{t+1}, \tilde{\lambda}_t) - \Psi(\tilde{\lambda}_t)\right)\right] \\
\leq{} & (1-\theta)\mathbb{E}\left(\Delta_t + \frac{\theta^2 v}{2\tau_1(1-\theta)} \|u_t - \lambda^*\|^2\right) + \left(\tau_1 + \frac{\theta}{\mu_\Psi}\right)\mathbb{E}\delta_t^2 - \left[\frac{\mu_\Psi \theta}{4} - \frac{\theta^2}{2\tau_1}(1-v))\right]\mathbb{E}\|\tilde{\lambda}_t - \lambda^*\|^2 \\
& - \frac{1}{4\tau_1}\mathbb{E}\|\lambda_{t+1} - \tilde{\lambda}_t\|^2 - \left[\frac{\theta^2}{2\tau_1} v(1-v)\left(\frac{1-\theta}{\beta\theta} - 1\right)^2 + \frac{\mu_\Psi(1-\theta)}{2}\right]\mathbb{E}\|\tilde{\lambda}_t - \lambda_t\|^2 \\
& + (1-\theta)\mathbb{E}(\nabla\Psi(\tilde{\lambda}_t) - \nabla_\lambda \mathcal{L}(X_{t+1}, \tilde{\lambda}_t))^\top (\lambda_t - \tilde{\lambda}_t) \\
& + p(1 - \mu_H \tau_2)^K \mathbb{E}\Bigg[ \left(1 + \frac{4\tau_1}{\mu_H}\right) B_{t-1} + \left(\frac{1}{4\tau_1} + \frac{a}{2\tau_1^2} + \frac{L_\Psi}{2}\right) \|\tilde{\lambda}_{t-1} - \lambda_t\|^2 + \frac{a}{2\tau_1^2} \|\lambda_{t+1} - \tilde{\lambda}_t\|^2 \\
& + \left(\frac{1}{a} - \frac{\mu_\Psi}{2}\right) \|\lambda_t - \tilde{\lambda}_t\|^2 + \left(\nabla\Psi(\tilde{\lambda}_t) - \nabla_\lambda \mathcal{L}(X_{t+1}, \tilde{\lambda}_{t+1})\right)^\top (\lambda_t - \tilde{\lambda}_t)\Bigg] + \frac{p L_H \tau_2^2 \sigma_H^2}{2} \sum_{k=0}^{K-1} (1 - \tau_2 \mu_H)^k \\
\leq{} & (1-\theta)\mathbb{E}\left(\Delta_t + \frac{\theta^2 v}{2\tau_1(1-\theta)} \|u_t - \lambda^*\|^2\right) + \left(\tau_1 + \frac{\theta}{\mu_\Psi}\right)\mathbb{E}\delta_t^2 - \left[\frac{\mu_\Psi \theta}{4} - \frac{\theta^2}{2\tau_1}(1-v))\right]\mathbb{E}\|\tilde{\lambda}_t - \lambda^*\|^2 \\
& - \left(\frac{1}{4\tau_1} - \frac{a p(1 - \mu_H \tau_2)^K}{2\tau_1^2}\right)\mathbb{E}\|\lambda_{t+1} - \tilde{\lambda}_t\|^2 \\
& - \left[\frac{\theta^2}{2\tau_1} v(1-v)\left(\frac{1-\theta}{\beta\theta} - 1\right)^2 + \frac{\mu_\Psi(1-\theta)}{2} - \left(\frac{1}{a} - \frac{\mu_\Psi}{2}\right) p(1 - \mu_H \tau_2)^K \right]\mathbb{E}\|\tilde{\lambda}_t - \lambda_t\|^2 \\
& + \left(1 - \theta + p(1 - \mu_H \tau_2)^K\right)\mathbb{E}(\nabla\Psi(\tilde{\lambda}_t) - \nabla_\lambda \mathcal{L}(X_{t+1}, \tilde{\lambda}_t))^\top (\lambda_t - \tilde{\lambda}_t) \\
& + p(1 - \mu_H \tau_2)^K \left(1 + \frac{4\tau_1}{\mu_H}\right)\mathbb{E}B_{t-1} + \left(\frac{1}{4\tau_1} + \frac{a}{2\tau_1^2} + \frac{L_\Psi}{2}\right) p(1 - \mu_H \tau_2)^K \mathbb{E}\|\tilde{\lambda}_{t-1} - \lambda_t\|^2 \\
& + \frac{p L_H \tau_2^2 \sigma_H^2}{2} \sum_{k=0}^{K-1} (1 - \tau_2 \mu_H)^k.
\end{aligned}
\tag{33}
$$

We want to choose $v$ (or equivalent $\beta$) such that the following holds,

$$
\frac{\mu_\Psi \theta}{4} - \frac{\theta^2}{2\tau_1}(1-v) \geq 0
\tag{A2}
$$

$$
\frac{\theta^2 v}{2\tau_1(1-\theta)} \leq \frac{\theta^2}{2\tau_1}.
\tag{B2}
$$

From (B2), we have

$$(B2) \Longleftrightarrow v \le 1 - \theta \Longleftrightarrow \beta \le \frac{1-\theta}{1+\theta}. \tag{34}$$

From (A2), we have

$$(A2) \Longleftrightarrow 1 - v \le \frac{\mu_\Psi \tau}{2\theta} \Longleftrightarrow v \ge 1 - \frac{\mu_\Psi \tau_1}{2\theta}. \tag{35}$$

From the upper and lower bounds for $v$ in (34) and (35), we require

$$1 - \frac{\mu_\Psi \tau_1}{2\theta} \le 1 - \theta \text{ and } 0 \le \theta \le 1 \Longleftrightarrow 0 \le \theta \le \sqrt{\frac{\mu_\Psi \tau_1}{2}}. \tag{36}$$

Similar to (20) and (21), we further pick $v$ (and therefore $\beta$) as the lower bound in (35):

$$\text{If } \theta \le \frac{\mu_\Psi \tau_1}{2}, \text{ then we choose } v = 0 \Longleftrightarrow \beta = 0; \tag{37}$$

$$\text{If } \theta > \frac{\mu_\Psi \tau_1}{2}, \text{ then we choose } v = 1 - \frac{\mu_\psi \tau_1}{2\theta} \Longleftrightarrow \beta = \frac{2\theta - \mu_\Psi \tau_1}{2\theta - \mu_\Psi \tau_1 \theta}(1 - \theta). \tag{38}$$

We denote $A_t = \Delta_t + \frac{\theta^2 v}{2\tau_1(1-\theta)}\|u_t - \lambda^*\|^2$. Since (37) and (38) hold, (33) becomes

$$\begin{aligned}
&\mathbb{E}\left[A_{t+1} + pB_t\right] \\
&\le (1-\theta)\mathbb{E}A_t + \left(\tau_1 + \frac{\theta}{\mu_\Psi}\right)\mathbb{E}\delta_t^2 - \left(\frac{1}{4\tau_1} - \frac{ap(1-\mu_H\tau_2)^K}{2\tau_1^2}\right)\mathbb{E}\|\lambda_{t+1} - \tilde{\lambda}_t\|^2 \\
&\quad - \left[\frac{\theta^2}{2\tau_1}v(1-v)\left(\frac{1-\theta}{\beta\theta} - 1\right)^2 + \frac{\mu_\Psi(1-\theta)}{2} - \left(\frac{1}{a} - \frac{\mu_\Psi}{2}\right)p(1-\mu_H\tau_2)^K\right]\mathbb{E}\|\tilde{\lambda}_t - \lambda_t\|^2 \\
&\quad + \left(1 - \theta + p(1-\mu_H\tau_2)^K\right)\mathbb{E}(\nabla\Psi(\tilde{\lambda}_t) - \nabla_\lambda\mathcal{L}(X_{t+1}, \tilde{\lambda}_t))^\top(\lambda_t - \tilde{\lambda}_t) \\
&\quad + p(1-\mu_H\tau_2)^K\left(1 + \frac{4\tau_1}{\mu_H}\right)\mathbb{E}B_{t-1} + \left(\frac{1}{4\tau_1} + \frac{a}{2\tau_1^2} + \frac{L_\Psi}{2}\right)p(1-\mu_H\tau_2)^K\mathbb{E}\|\tilde{\lambda}_{t-1} - \lambda_t\|^2 \\
&\quad + \frac{pL_H\tau_2^2\sigma_H^2}{2}\sum_{k=0}^{K-1}(1-\tau_2\mu_H)^k.
\end{aligned} \tag{39}$$

In the first case (37), when we choose $v = \beta = 0$, we have $\lambda_t = \tilde{\lambda}_t$ and (39) becomes

$$\begin{aligned}
&\mathbb{E}\left[A_{t+1} + pB_t\right] \\
&\le (1-\theta)\mathbb{E}A_t + \left(\tau_1 + \frac{\theta}{\mu_\Psi}\right)\mathbb{E}\delta_t^2 - \left(\frac{1}{4\tau_1} - \frac{ap(1-\mu_H\tau_2)^K}{2\tau_1^2}\right)\mathbb{E}\|\lambda_{t+1} - \lambda_t\|^2 \\
&\quad + p(1-\mu_H\tau_2)^K\left(1 + \frac{4\tau_1}{\mu_H}\right)\mathbb{E}B_{t-1} + \left(\frac{1}{4\tau_1} + \frac{a}{2\tau_1^2} + \frac{L_\Psi}{2}\right)p(1-\mu_H\tau_2)^K\mathbb{E}\|\lambda_{t-1} - \lambda_t\|^2 \\
&\quad + \frac{pL_H\tau_2^2\sigma_H^2}{2}\sum_{k=0}^{K-1}(1-\tau_2\mu_H)^k.
\end{aligned}$$

Note that $\delta_t^2 \le \frac{4}{\mu_H}B_t$ and $\theta \le \frac{\mu_\Psi \tau_1}{2}$. We pick $a = \tau_1/2$ and denote $c = (1-\mu_H\tau_2)^K$.

$$\begin{aligned}
&\mathbb{E}\left[A_{t+1} + pB_t\right] \\
&\le (1-\theta)\mathbb{E}A_t + \frac{6\tau_1}{\mu_H}\mathbb{E}B_t - \left(\frac{1}{4\tau_1} - \frac{pc}{4\tau_1}\right)\mathbb{E}\|\lambda_{t+1} - \lambda_t\|^2 + pc\left(1 + \frac{4\tau_1}{\mu_H}\right)\mathbb{E}B_{t-1} \\
&\quad + \frac{5pc}{8\tau_1}\mathbb{E}\|\lambda_{t-1} - \lambda_t\|^2 + \frac{pL_H\tau_2^2\sigma_H^2}{2}\sum_{k=0}^{K-1}(1-\tau_2\mu_H)^k.
\end{aligned} \tag{40}$$

In the second case (38), by our choice of $\upsilon$ and $\beta$, similar to (23), (39) becomes

$$\mathbb{E}\left[A_{t+1} + pB_t\right]$$
$$\leq (1-\theta)\mathbb{E}A_t + \frac{12\theta}{\mu_H\mu_\Psi}\mathbb{E}B_t - \left(\frac{1}{4\tau_1} - \frac{pc}{4\tau_1}\right)\mathbb{E}\|\lambda_{t+1} - \tilde{\lambda}_t\|^2$$
$$- \left[\frac{\mu_\Psi(1-\theta)^2}{2(2\theta - \mu_\Psi\tau_1)} + \frac{\mu_\Psi(1-\theta)}{2} - \left(\frac{2}{\tau_1} - \frac{\mu_\Psi}{2}\right)pc\right]\mathbb{E}\|\tilde{\lambda}_t - \lambda_t\|^2$$
$$+ (1 - \theta + pc)\,\mathbb{E}(\nabla\Psi(\tilde{\lambda}_t) - \nabla_\lambda\mathcal{L}(X_{t+1}, \tilde{\lambda}_t))^\top(\lambda_t - \tilde{\lambda}_t)$$
$$+ pc\left(1 + \frac{4\tau_1}{\mu_H}\right)\mathbb{E}B_{t-1} + \frac{5pc}{8\tau_1}\mathbb{E}\|\tilde{\lambda}_{t-1} - \lambda_t\|^2 + \frac{pL_H\tau_2^2\sigma_H^2}{2}\sum_{k=0}^{K-1}(1 - \tau_2\mu_H)^k. \tag{41}$$

**Part IV:** Convergence. We pick $\theta, p, \tau_1$ and $\tau_2$ in (40) and (41) to show convergence.

We simplify (41), with $pc \leq 1/6$, we have

$$\mathbb{E}\left[A_{t+1} + pB_t + \frac{5pc}{4\tau_1}\|\lambda_{t+1} - \tilde{\lambda}_t\|^2\right]$$
$$\leq (1-\theta)\mathbb{E}A_t + \frac{12\theta}{\mu_H\mu_\Psi}\mathbb{E}B_t - \left[\frac{\mu_\Psi(1-\theta)^2}{2(2\theta - \mu_\Psi\tau_1)} - \frac{2pc}{\tau_1}\right]\mathbb{E}\|\tilde{\lambda}_t - \lambda_t\|^2$$
$$+ (1 - \theta + pc)\,\mathbb{E}(\nabla\Psi(\tilde{\lambda}_t) - \nabla_\lambda\mathcal{L}(X_{t+1}, \tilde{\lambda}_t))^\top(\lambda_t - \tilde{\lambda}_t)$$
$$+ pc\left(1 + \frac{4\tau_1}{\mu_H}\right)\mathbb{E}B_{t-1} + \frac{5pc}{8\tau_1}\mathbb{E}\|\tilde{\lambda}_{t-1} - \lambda_t\|^2 + \frac{pL_H\tau_2^2\sigma_H^2}{2}\sum_{k=0}^{K-1}(1 - \tau_2\mu_H)^k. \tag{42}$$

In order to make the coefficient for $\mathbb{E}\|\tilde{\lambda}_t - \lambda_t\|^2$ to be negative on the right side, we choose $\theta$ such that

$$\frac{2pc}{\tau_1} \leq \frac{\mu_\Psi(1-\theta)^2}{4(2\theta - \mu_\Psi\tau_1)},$$

which is always possible because as we pick $\theta \to \mu_\Psi\tau_1/2$, the right side goes to infinity. Consider the third and fourth terms on the right side of (42),

$$-\left[\frac{\mu_\Psi(1-\theta)^2}{2(2\theta - \mu_\Psi\tau_1)} - \frac{2pc}{\tau_1}\right]\mathbb{E}\|\tilde{\lambda}_t - \lambda_t\|^2 + (1 - \theta + pc)\,\mathbb{E}(\nabla\Psi(\tilde{\lambda}_t) - \nabla_\lambda\mathcal{L}(X_{t+1}, \tilde{\lambda}_t))^\top(\lambda_t - \tilde{\lambda}_t)$$
$$\leq \frac{\mu_\Psi(1-\theta)^2}{4(2\theta - \mu_\Psi\tau_1)}\mathbb{E}\|\tilde{\lambda}_t - \lambda_t\|^2 + (1 - \theta + pc)\,\mathbb{E}(\nabla\Psi(\tilde{\lambda}_t) - \nabla_\lambda\mathcal{L}(X_{t+1}, \tilde{\lambda}_t))^\top(\lambda_t - \tilde{\lambda}_t)$$
$$\leq \frac{(2\theta - \mu_\Psi\tau_1)(1 - \theta + pc)^2}{\mu_\Psi(1-\theta)^2}\mathbb{E}\|\nabla\Psi(\tilde{\lambda}_t) - \nabla_\lambda\mathcal{L}(X_{t+1}, \tilde{\lambda}_t)\|^2 \leq \frac{4(2\theta - \mu_\Psi\tau_1)(1 - \theta + pc)^2}{\mu_H\mu_\Psi(1-\theta)^2}\mathbb{E}B_t.$$

Plugging it back into (42),

$$\mathbb{E}\left[A_{t+1} + \left(p - \frac{12\theta}{\mu_H\mu_\Psi} - \frac{4(2\theta - \mu_\Psi\tau_1)(1 - \theta + pc)^2}{\mu_H\mu_\Psi(1-\theta)^2}\right)B_t + \frac{5pc}{4\tau_1}\|\lambda_{t+1} - \tilde{\lambda}_t\|^2\right] \tag{43}$$
$$\leq (1-\theta)\mathbb{E}A_t + pc\left(1 + \frac{4\tau_1}{\mu_H}\right)\mathbb{E}B_{t-1} + \frac{5pc}{8\tau_1}\mathbb{E}\|\tilde{\lambda}_{t-1} - \lambda_t\|^2 + \frac{pL_H\tau_2^2\sigma_H^2}{2}\sum_{k=0}^{K-1}(1 - \tau_2\mu_H)^k.$$

For convenience, we denote the coefficients for $B_t$ and $B_{t-1}$ as $m_1$ and $m_2$:

$$m_1 = p - \frac{12\theta}{\mu_H\mu_\Psi} - \frac{4(2\theta - \mu_\Psi\tau_1)(1 - \theta + pc)^2}{\mu_H\mu_\Psi(1-\theta)^2},$$
$$m_2 = pc\left(1 + \frac{4\tau_1}{\mu_H}\right)$$

We need to guarantee $m_1 > 0$. Note that

$$
\begin{aligned}
m_1 &= p - \frac{12\theta}{\mu_H \mu_\Psi} - \frac{4(2\theta - \mu_\Psi \tau_1)(1 - \theta + pc)^2}{\mu_H \mu_\Psi (1 - \theta)^2} \\
&= p - \frac{1}{\mu_H \mu_\Psi} \left[ 12\theta + \frac{4(2\theta - \mu_\Psi \tau_1)(1 - \theta + pc)^2}{(1 - \theta)^2} \right] \\
&\geq p - \frac{1}{\mu_H \mu_\Psi} \left[ 12\theta + 8(2\theta - \mu_\Psi \tau_1) \right] \geq p \left[ 1 - \frac{28\theta}{\mu_H \mu_\Psi p} \right] \triangleq \tilde{m}_1,
\end{aligned}
\tag{44}
$$

where in the first inequality, we use $\theta \leq 1/2$ and $pc \leq 1/6$. We then require $\theta < \frac{\mu_H \mu_\Psi p}{28}$ to make sure $m_1 > 0$. (43) implies a contraction rate of $\max\{1 - \theta, m_2/\tilde{m}_1, 1/2\}$ up to the last noise term:

$$
\mathbb{E}\left[ A_{t+1} + p\left(1 - \frac{28\theta}{\mu_H \mu_\Psi p}\right) B_t + \frac{5pc}{4\tau_1} \|\lambda_{t+1} - \tilde{\lambda}_t\|^2 \right]
\tag{45}
$$

$$
\leq \max\left\{ 1 - \theta, \frac{c\left(1 + \frac{4\tau_1}{\mu_H}\right)}{1 - \frac{28\theta}{\mu_H \mu_\Psi p}}, \frac{1}{2} \right\} \mathbb{E}\left[ A_t + p\left(1 - \frac{28\theta}{\mu_H \mu_\Psi p}\right) B_{t-1} + \frac{5pc}{4\tau_1} \mathbb{E}\|\tilde{\lambda}_{t-1} - \lambda_t\|^2 \right]
$$

$$
+ \frac{pL_H \tau_2^2 \sigma_H^2}{2} \sum_{k=0}^{K-1} (1 - \tau_2 \mu_H)^k.
\tag{46}
$$

Throughout the proof, we imposed the following conditions on the algorithm parameters:

$$
\tau_1 \leq \frac{1}{4L_\Psi}, \ \tau_2 \leq \frac{1}{L_H}, \ \sqrt{\frac{\mu_\Psi \tau_1}{2}} \geq \theta > \frac{\mu_\Psi \tau_1}{2},
\tag{47}
$$

$$
pc \leq \frac{1}{6},
\tag{48}
$$

$$
\frac{2pc}{\tau_1} \leq \frac{\mu_\Psi (1 - \theta)^2}{4(2\theta - \mu_\Psi \tau_1)} \iff 2\theta - \mu_\Psi \tau_1 \leq \frac{\tau_1 \mu_\Psi (1 - \theta)^2}{8pc} \overset{\theta \leq 1/2}{\Longleftarrow} 2\theta - \mu_\Psi \tau_1 \leq \frac{\tau_1 \mu_\Psi}{32pc},
\tag{49}
$$

$$
\theta < \frac{\mu_H \mu_\Psi p}{28}.
\tag{50}
$$

$\square$

We restate Theorem 2.4.

**Theorem C.2.** *Suppose Assumptions 2.1 and 2.2 hold. Consider Algorithm 1 with suitable stepsizes $\tau_1, \tau_2$ and momentum parameter $\beta > 0$. Then, for any number of local steps $K \geq 1$, Algorithm 1 finds an $\epsilon$-accurate solution within*

$$
\tilde{\mathcal{O}}\left( \max\left\{ \frac{\kappa^4 \sigma^2}{\mu^2 (1 - \sigma_2)^3 \epsilon K}, \frac{\kappa^2}{K(1 - \sigma_2)^2}, \frac{\kappa}{1 - \sigma_2} \right\} \right)
$$

*communication rounds.*

*Moreover, if*

$$
K \geq \tilde{\Theta}\left( \max\left\{ \frac{\kappa^4 \sigma^2}{\mu^2 (1 - \sigma_2) \epsilon}, \kappa \right\} \right),
$$

*then, for any $\alpha \in [1/2, 1]$, Algorithm 1 achieves the communication complexity*

$$
\tilde{\mathcal{O}}\left( \left( \frac{\kappa}{1 - \sigma_2} \right)^\alpha \right),
$$

*provided that the momentum parameter is chosen as $\beta = 1 - \Theta\left((\mu \tau_1)^{1-\alpha}\right) > 0$. In particular, choosing $\alpha = 1/2$ recovers the near-optimal communication complexity $\tilde{\mathcal{O}}\left( \sqrt{\frac{\kappa}{1-\sigma_2}} \right).$*

*Proof.* We choose the parameters $\tau_1$, $\tau_2$, $\theta$, and $p$ according to the local step $K$ such that conditions (47)–(50) are satisfied, and then establish the resulting convergence complexity. Define

$$q = \max\left\{1 - \theta, \frac{c\left(1 + \frac{4\tau_1}{\mu_H}\right)}{1 - \frac{28\theta}{\mu_H\mu_\Psi p}}, \frac{1}{2}\right\}, \quad P_t = \mathbb{E}\left[A_t + p\left(1 - \frac{28\theta}{\mu_H\mu_\Psi p}\right)B_{t-1} + \frac{5pc}{4\tau_1}\|\lambda_t - \tilde{\lambda}_{t-1}\|^2\right].$$

By (45), we have

$$P_{t+1} \leq qP_t + \frac{p(1-c)L_H\sigma_H^2\tau_2}{2\mu_H}.$$

Recursively applying this inequality yields

$$P_T \leq q^T P_0 + \frac{p(1-c)L_H\sigma_H^2\tau_2}{2\mu_H(1-q)}. \tag{51}$$

We now consider two choices of $\theta$: $\theta = \Theta(\mu_\Psi\tau_1)$ and $\theta = \Theta((\mu_\Psi\tau_1)^\alpha)$ for $1/2 \leq \alpha < 1$.

**Case 1: $\theta = \Theta(\mu_\Psi\tau_1)$.** We choose $\theta = \frac{2\mu_\Psi\tau_1}{3}$. With this choice, the requirements on $\tau_1$ and $\theta$ in (47) are satisfied provided that $\tau_1 \leq \frac{1}{4L_\Psi}$. We structure the proof into several steps.

*Step 1.* We want $q$ to take the value $1 - \theta$. To this end, it suffices to require

$$1 - \theta \geq \frac{c\left(1 + \frac{4\tau_1}{\mu_H}\right)}{1 - \frac{28\theta}{\mu_H\mu_\Psi p}}, \quad 1 - \theta \geq \frac{1}{2}. \tag{52}$$

*Step 2.* Since $q = 1 - \theta$,

$$P_T \leq (1-\theta)^T P_0 + \frac{p(1-c)L_H\sigma_H^2\tau_2}{2\mu_H\theta}. \tag{53}$$

First, we want the noise term to be smaller than $\epsilon/2$. Note that

$$1 - c = 1 - (1 - \mu_H\tau_2)^K \leq K\mu_H\tau_2.$$

Plugging it back to the second term in (53), we can choose $\tau_2$ satisfying

$$\frac{pKL_H\sigma_H^2\tau_2^2}{2\theta} \leq \frac{\epsilon}{2} \quad \Longleftarrow \quad \tau_2 = \min\left\{\sqrt{\frac{\epsilon\theta}{pKL_H\sigma_H^2}}, \frac{1}{L_H}\right\}. \tag{54}$$

*Step 3.* Now we let $T$ large enough so that the contraction term in (53) is small enough:

$$(1-\theta)^T P_0 \leq \frac{\epsilon}{2} \quad \Longleftarrow \quad T = \mathcal{O}\left(\frac{1}{\theta}\log\frac{P_0}{\epsilon}\right).$$

*Step 4.* We want (52) to hold. Since $c = (1 - \mu_H\tau_2)^K \leq \exp\{-K\mu_H\tau_2\}$, we let $\tau_2$ no less than

$$\frac{1}{K\mu_H}\log\frac{\left(1 + \frac{4\tau_1}{\mu_H}\right)}{(1-\theta)\left(1 - \frac{28\theta}{\mu_H\mu_\Psi p}\right)} \leq \frac{1}{K\mu_H}\left(\frac{4\tau_1}{\mu_H} + 2\theta + \frac{56\theta}{\mu_H\mu_\Psi p}\right)$$

where we use $\log(1+u) \leq u$ and $-\log(1-u) \leq 2u$. Therefore, we require

$$\tau_2 \geq \frac{2\theta}{K\mu_H}\left(\frac{4}{\mu_\Psi\mu_H} + \frac{28}{\mu_H\mu_\Psi p}\right)$$

Combining with (54),

$$\sqrt{\frac{\epsilon\theta}{pKL_H\sigma_H^2}} \geq \frac{2\theta}{K\mu_H}\left(\frac{4}{\mu_\Psi\mu_H} + \frac{28}{\mu_H\mu_\Psi p}\right), \quad \frac{1}{L_H} \geq \frac{2\theta}{K\mu_H}\left(\frac{4}{\mu_\Psi\mu_H} + \frac{28}{\mu_H\mu_\Psi p}\right)$$

$$\Longleftarrow \quad \theta \leq \min\left\{\frac{p\mu_H^2(\mu_\Psi\mu_H)^2}{64^2 L_H\sigma_H^2}\cdot\epsilon K, \frac{K\mu_H(\mu_\Psi\mu_H)p}{64 L_H}\right\}. \tag{55}$$

Further with (47), we can choose

$$\theta = \min\left\{\frac{p\mu_H^2(\mu_\Psi\mu_H)^2}{64^2 L_H\sigma_H^2}\cdot\epsilon K, \frac{K\mu_H(\mu_\Psi\mu_H)p}{64 L_H}, \frac{\mu_\Psi}{160 L_\Psi}\right\}. \tag{56}$$

By choosing $p = 3/32$, the requirements (49) and (50) are satisfied. Then the outer loop (communication) complexity is

$$T = \mathcal{O}\left(\frac{1}{\theta}\log\frac{P_0}{\epsilon}\right) = \widetilde{\mathcal{O}}\left(\max\left\{\frac{L_H\sigma_H^2}{\mu_H^2(\mu_\Psi\mu_H)^2(\epsilon K)}, \frac{L_H}{K\mu_H(\mu_\Psi\mu_H)}, \frac{L_\Psi}{\mu_\Psi}\right\}\right).$$

The total gradient complexity is

$$K\cdot T = \widetilde{\mathcal{O}}\left(\max\left\{\frac{L_H\sigma_H^2}{\mu_H^2(\mu_\Psi\mu_H)^2\epsilon}, \frac{L_H}{\mu_H(\mu_\Psi\mu_H)}, \frac{KL_\Psi}{\mu_\Psi}\right\}\right).$$

This is complexity for $P_T \leq \epsilon$. Our goal is to find $\mathbb{E}\|\bar{x}^T - x^*\|^2 \leq \epsilon$, so we establish a relationship between them. By Jensen's inequality and definition of $P_t$,

$$\mathbb{E}\|\bar{x}^T - x^*\|^2 \leq \frac{1}{M}\sum_{m=1}^M \mathbb{E}\|x_m^{T,K} - x^*\|^2 \leq \frac{2\mathbb{E}\|X^{T,K} - X^*(\lambda^T)\|^2}{M} + \frac{4M}{\mu^2}\mathbb{E}\|\lambda^T - \lambda^*\|^2$$

$$\leq \frac{4\mathbb{E}B_T}{\mu_H M} + \frac{8M}{\mu^2\mu_\Psi}\mathbb{E}\Delta_t = \frac{4\mathbb{E}B_T}{\mu} + \frac{8L}{\mu^2(1-\sigma_2)}\mathbb{E}\Delta_t$$

$$\leq \frac{8L}{\mu^2(1-\sigma_2)}P_T + \frac{4}{\mu}\cdot\frac{1}{p\left(1 - \frac{28\theta}{\mu_\Psi\mu_H p}\right)}P_{T+1} \leq \left(\frac{8L}{\mu^2(1-\sigma_2)} + \frac{640}{\mu}\right)P_{T+1}, \tag{57}$$

where in the last inequality we use our choice of $p$ and $P_{T+1} < P_T$. Note that $\mu_H = \mu/M$, $L_H = L/M$, $L_\Psi = \frac{2M}{\mu}$ and $\mu_\Psi = \frac{M(1-\sigma_2)}{L}$. The final communication complexity for $\mathbb{E}\|\bar{x}^T - x^*\|^2 \leq \epsilon$ is

$$T = \widetilde{\mathcal{O}}\left(\max\left\{\frac{\kappa^4\sigma^2}{\mu^2(1-\sigma_2)^3(\epsilon K)}, \frac{\kappa^2}{K(1-\sigma_2)^2}, \frac{\kappa}{1-\sigma_2}\right\}\right). \tag{58}$$

**Case 2: $\theta = \Theta((\mu_\Psi\tau_1)^\alpha)$ with $\frac{1}{2} \leq \alpha < 1$.** We choose $\theta = \left(\frac{\mu_\Psi\tau_1}{2}\right)^\alpha$. Since $\tau_1 \leq \frac{1}{L_\Psi}$, it follows that $\theta \leq \left(\frac{\mu_\Psi}{2L_\Psi}\right)^\alpha$. We follow the same steps as in Case 1, namely *Steps 1–3*, with this choice of $\theta$.

*Step 4.* We need to choose $p$ so that (48), (49) and (50) are satisfied. From (49), we want

$$2\theta - \mu_\Psi\tau_1 \leq \frac{\tau_1\mu_\Psi}{32pc} \quad \Longleftarrow \quad \theta \leq \frac{\tau_1\mu_\Psi}{64pc} = \frac{\theta^{\frac{1}{\alpha}}}{32pc} \quad \Longleftarrow \quad p \leq \frac{\theta^{\frac{1-\alpha}{\alpha}}}{32c}. \tag{59}$$

To satisfy (50),

$$\theta < \frac{\mu_H\mu_\Psi p}{28} \quad \Longleftrightarrow \quad p > \frac{28\theta}{\mu_H\mu_\Psi}. \tag{60}$$

In order for (59) and (60) to hold at the same time, we need

$$\frac{28\theta}{\mu_H\mu_\Psi} < \frac{\theta^{\frac{1-\alpha}{\alpha}}}{32c} \quad \Longleftarrow \quad c \leq \frac{\mu_H\mu_\Psi}{1000}\cdot\theta^{\frac{1-2\alpha}{\alpha}}. \tag{61}$$

When $\alpha > \frac{1}{2}$, (61) holds if

$$\text{either} \quad \theta \leq \left(\frac{\mu_H \mu_\Psi}{1000}\right)^{\frac{\alpha}{2\alpha-1}} \quad \text{for arbitrary } K \geq 1,$$

$$\text{or} \quad K \text{ is sufficiently large such that } c \leq \frac{\mu_H \mu_\Psi}{1000} \theta^{\frac{1-2\alpha}{\alpha}}. \tag{62}$$

When $\alpha = \frac{1}{2}$, (61) holds if

$$K \text{ is sufficiently large such that } c \leq \frac{\mu_H \mu_\Psi}{1000} \tag{63}$$

*Step 5(a): When $\alpha > \frac{1}{2}$ with arbitrary $K$.* We want (52) to hold, so we let $\tau_2$ no less than

$$\frac{1}{K\mu_H} \log \frac{\left(1 + \frac{4\tau_1}{\mu_H}\right)}{(1-\theta)\left(1 - \frac{28\theta}{\mu_H \mu_\Psi p}\right)} \leq \frac{1}{K\mu_H}\left(\frac{4\tau_1}{\mu_H} + 2\theta + \frac{56\theta}{\mu_H \mu_\Psi p}\right)$$

where we use $\log(1+u) \leq u$ and $-\log(1-u) \leq 2u$. Therefore, as $p < 1$ and $\theta^{\frac{1}{\alpha}} \leq \theta$, we require,

$$\tau_2 \geq \frac{66\theta}{K\mu_H(\mu_H\mu_\Psi)p}. \tag{64}$$

Combining with (54), by choosing $p \leq \frac{\theta^{\frac{1-\alpha}{\alpha}}}{32c}$ satisfying (62),

$$\sqrt{\frac{\epsilon\theta}{pKL_H\sigma_H^2}} \geq \frac{66\theta}{K\mu_H(\mu_H\mu_\Psi)p}, \quad \frac{1}{L_H} \geq \frac{66\theta}{K\mu_H(\mu_H\mu_\Psi)p}$$

$$\Longleftarrow \quad \theta \leq \min\left\{ \left[\frac{\mu_H^2(\mu_\Psi\mu_H)^2}{32 \times 66^2 L_H \sigma_H^2} \cdot \epsilon K\right]^{\frac{\alpha}{2\alpha-1}}, \left[\frac{K\mu_H(\mu_\Psi\mu_H)^2 p}{66 L_H}\right]^{\frac{\alpha}{2\alpha-1}} \right\}. \tag{65}$$

Further with (47), we choose

$$\theta = \min\left\{ \left[\frac{\mu_H^2(\mu_\Psi\mu_H)^2}{32 \times 66^2 L_H \sigma_H^2} \cdot \epsilon K\right]^{\frac{\alpha}{2\alpha-1}}, \left[\frac{K\mu_H(\mu_\Psi\mu_H)^2 p}{66 L_H}\right]^{\frac{\alpha}{2\alpha-1}}, \left(\frac{\mu_\Psi}{8L_\Psi}\right)^\alpha, \left(\frac{\mu_H\mu_\Psi}{1000}\right)^{\frac{\alpha}{2\alpha-1}} \right\}. \tag{66}$$

Then the outer loop (communication) complexity is

$$T = \mathcal{O}\left(\frac{1}{\theta}\log\frac{P_0}{\epsilon}\right) = \widetilde{\mathcal{O}}\left(\max\left\{ \left[\frac{L_H\sigma_H^2}{\mu_H^2(\mu_\Psi\mu_H)^2(\epsilon K)}\right]^{\frac{\alpha}{2\alpha-1}}, \left[\frac{L_H}{K\mu_H(\mu_\Psi\mu_H)}\right]^{\frac{\alpha}{2\alpha-1}}, \left(\frac{L_\Psi}{\mu_\Psi}\right)^\alpha, \left(\frac{1000}{\mu_H\mu_\Psi}\right)^{\frac{\alpha}{2\alpha-1}} \right\}\right).$$

The total gradient complexity is

$$K \cdot T = \widetilde{\mathcal{O}}\left(\max\left\{ \left[\frac{L_H\sigma_H^2}{\mu_H^2(\mu_\Psi\mu_H)^2\epsilon}\right]^{\frac{\alpha}{2\alpha-1}} K^{\frac{\alpha-1}{2\alpha-1}}, K\left[\frac{L_H}{\mu_H(\mu_\Psi\mu_H)}\right]^{\frac{\alpha}{2\alpha-1}}, K\left(\frac{L_\Psi}{\mu_\Psi}\right)^\alpha, K\left(\frac{1000}{\mu_H\mu_\Psi}\right)^{\frac{\alpha}{2\alpha-1}} \right\}\right).$$

*Step 5 (b): When $\alpha \geq \frac{1}{2}$ with $K$ large enough.* From (62) and (63), we require

$$c = (1 - \mu_H\tau_2)^K \leq \frac{\mu_H\mu_\Psi}{1000} \theta^{\frac{1-2\alpha}{\alpha}}$$

Since $c = (1 - \mu_H\tau_2)^K \leq \exp\{-K\mu_H\tau_2\}$, this can be guaranteed by choosing

$$K\tau_2 \geq \frac{1}{\mu_H} \log \frac{1000}{\mu_H\mu_\Psi}. \tag{67}$$

We still choose $\tau_2$ as (54), and we need $K$ large enough so that $\tau_2$ satisfies (64) and (67). Here we pick $p = \frac{30\theta}{\mu_H \mu_\Psi}$. In order for $\tau_2$ to satisfy (64),

$$\sqrt{\frac{\epsilon\theta}{pKL_H\sigma_H^2}} \geq \frac{66\theta}{K\mu_H(\mu_H\mu_\Psi)p}, \quad \frac{1}{L_H} \geq \frac{66\theta}{K\mu_H(\mu_H\mu_\Psi)p}$$

$$\Longleftarrow \quad K \geq \max\left\{\frac{150L_H\sigma_H^2}{\mu_H^2(\mu_H\mu_\Psi)\epsilon}, \frac{33L_H}{15\mu_H}\right\}. \tag{68}$$

In order for $\tau_2$ to satisfy (67),

$$K\sqrt{\frac{\epsilon\theta}{pKL_H\sigma_H^2}} \geq \frac{1}{\mu_H}\log\frac{1000}{\mu_H\mu_\Psi}, \quad \frac{K}{L_H} \geq \frac{1}{\mu_H}\log\frac{1000}{\mu_H\mu_\Psi}$$

$$\Longleftarrow \quad K \geq \max\left\{\frac{30L_H\sigma_H^2}{\mu_H^2(\mu_H\mu_\Psi)\epsilon}\log^2\frac{1000}{\mu_H\mu_\Psi}, \frac{L_H}{\mu_H}\log\frac{1000}{\mu_H\mu_\Psi}\right\}. \tag{69}$$

Therefore, these requirements for $\tau_2$ can be satisfied when

$$K \geq \max\left\{\frac{150L_H\sigma_H^2}{\mu_H^2(\mu_H\mu_\Psi)\epsilon}\log^2\frac{1000}{\mu_H\mu_\Psi}, \frac{3L_H}{\mu_H}\log\frac{1000}{\mu_H\mu_\Psi}\right\}. \tag{70}$$

Now we can choose $\theta = \left(\frac{\mu_\Psi}{8L_\Psi}\right)^\alpha$. Then the outer loop (communication) complexity is

$$T = \mathcal{O}\left(\frac{1}{\theta}\log\frac{P_0}{\epsilon}\right) = \widetilde{\mathcal{O}}\left(\left(\frac{L_\Psi}{\mu_\Psi}\right)^\alpha\right).$$

The total gradient complexity is

$$K \cdot T = \widetilde{\mathcal{O}}\left(K\left(\frac{L_\Psi}{\mu_\Psi}\right)^\alpha\right).$$

This is complexity for $P_T \leq \epsilon$. To find $\mathbb{E}\|\bar{x}^T - x^*\|^2 \leq \epsilon$, by (57),

$$\mathbb{E}\|\bar{x}^T - x^*\|^2 \leq \frac{8L}{\mu^2(1-\sigma_2)}P_T + \frac{4}{\mu} \cdot \frac{1}{p\left(1 - \frac{28\theta}{\mu_\Psi\mu_H p}\right)}P_{T+1}$$

$$\leq \left(\frac{8L}{\mu^2(1-\sigma_2)} + \frac{2(1-\sigma_2)}{\mu\kappa\theta}\right)P_{T+1} = \left(\frac{8L}{\mu^2(1-\sigma_2)} + \frac{2^{1+\alpha}(1-\sigma_2)^{1-\alpha}}{\mu\kappa^{1-\alpha}}\right)P_{T+1}$$

Therefore, as long as

$$K \geq \widetilde{\Theta}\left(\max\left\{\frac{\kappa^4\sigma^2}{\mu^2(1-\sigma_2)\epsilon}, \kappa\right\}\right), \tag{71}$$

we have communication complexity $\widetilde{\mathcal{O}}\left(\left(\frac{\kappa}{1-\sigma_2}\right)^\alpha\right)$.

$\square$

*Remark* C.3. In the case $\theta = \frac{2\mu_\Psi\tau_1}{3}$, the algorithm employs a relatively small momentum parameter. According to (58), when the number of local steps satisfies $K \leq \mathcal{O}(\epsilon^{-1})$, the communication complexity is $\widetilde{\mathcal{O}}(1/(\epsilon K))$, while the gradient complexity is $\widetilde{\mathcal{O}}(1/\epsilon)$. When the number of local steps is sufficiently large, namely $K \geq \widetilde{\Theta}\left(\max\{\sigma^2/\epsilon, \kappa/(1-\sigma_2)\}\right)$, the first two terms are dominated by the third term in (58), so the communication complexity is $\widetilde{\mathcal{O}}(\kappa/(1-\sigma_2))$.

*Remark* C.4. In the case $\theta = \left(\frac{\mu_\Psi\tau_1}{2}\right)^\alpha$ with $\frac{1}{2} \leq \alpha < 1$, when the number of local steps satisfies $K \geq \widetilde{\Theta}\left(\max\{\sigma^2/\epsilon, \kappa/(1-\sigma_2)\}\right)$, the resulting communication complexity is $\widetilde{\mathcal{O}}((\kappa/(1-\sigma_2))^\alpha)$. Since the momentum parameter $\beta$ depends on $\theta$, choosing a smaller $\alpha$ corresponds to using a larger momentum, which in turn leads to improved communication complexity.

---

**Algorithm 4** Decentralized Catalyst

---

1: **Input:** initial point $x^0$. Set $y^0 = x^0$. In the strongly convex setting, set $q = \frac{\mu}{\mu+2L}$ and $\alpha_1 = \sqrt{q}$; in the convex and nonconvex settings, set $q = 0$ and $\alpha_1 = 1$.

2: **for** $s = 1, 2, \ldots, S$ **do**

3:    Approximately solve the decentralized subproblem

$$\min_z F^s(z) := \frac{1}{M} \sum_{m=1}^{M} F_m^s(z), \qquad F_m^s(z) := F_m(z) + L\|z - y_m^s\|^2,$$

   and obtain local iterates $(x_1^s, \ldots, x_M^s)$ satisfying

$$\mathbb{E}\big[F^s(\bar{x}^s) - \min_z F^s(z)\big] \le \epsilon^s, \qquad \bar{x}^s = \frac{1}{M} \sum_{m=1}^{M} x_m^s.$$

4:    Update the momentum parameter. In the strongly convex and convex settings, choose $\alpha_s$ and $\beta_s$ such that

$$\alpha_s^2 = (1 - \alpha_s)\alpha_{s-1}^2 + q\alpha_s, \qquad \beta_s = \frac{\alpha_{s-1}(1 - \alpha_{s-1})}{\alpha_{s-1}^2 + \alpha_s}.$$

   In the nonconvex setting, set $\beta_s = 0$.

5:    Update the prox centers:
$$y_m^{s+1} = x_m^s + \beta_s(x_m^s - x_m^{s-1}), \qquad m = 1, \ldots, M.$$

6: **end for**

---

*Remark* C.5. The term $\|\lambda_{t+1} - \tilde{\lambda}_t\|^2$ can be interpreted as a proxy for the network-wide consensus error. Indeed, by Lemma 3 in (Li et al., 2020),

$$\frac{1}{M} \sum_{m=1}^{M} \big\|X_{t,K}^m - \overline{X}_{t,K}\big\|^2 \le \frac{2}{1-\sigma_2} \|UX_{t,K}\|^2 = \frac{2}{(1-\sigma_2)\tau_1^2} \|\lambda_{t+1} - \tilde{\lambda}_t\|^2,$$

where $\overline{X}_{t,K} := \frac{1}{M} \sum_{m=1}^{M} X_{t,K}^m$. Therefore, convergence of the outer loop (in particular, $\|\lambda_{t+1} - \tilde{\lambda}_t\| \to 0$) also implies the convergence of the consensus criterion.

## D. Extension to Non-strongly Convex Settings

**Decentralized Catalyst.** Li & Lin (2020) adapted Catalyst to the decentralized setting; see Algorithm 4. At the $s$-th outer iteration, the method approximately solves the decentralized subproblem

$$\min_x F^s(x) := \frac{1}{M} \sum_{m=1}^{M} F_m^s(x), \qquad F_m^s(x) := F_m(x) + L\|x - y_m^s\|^2. \tag{72}$$

Let $\bar{y}^s = \frac{1}{M} \sum_{m=1}^{M} y_m^s$. Then

$$F^s(x) = F(x) + L\|x - \bar{y}^s\|^2 + \frac{L}{M} \sum_{m=1}^{M} \|y_m^s\|^2 - L\|\bar{y}^s\|^2.$$

Thus, up to an additive constant independent of $x$, the subproblem is equivalent to

$$\min_x G^s(x) := F(x) + L\|x - \bar{y}^s\|^2.$$

Consequently, the criterion

$$\mathbb{E}\big[F^s(\bar{x}^s) - \min_x F^s(x)\big] \le \epsilon^s, \qquad \bar{x}^s = \frac{1}{M} \sum_{m=1}^{M} x_m^s,$$

---

**Algorithm 5** Catalyst (Lin et al., 2018; Paquette et al., 2018)

---

1: **Input:** initial point $x^0$. Set $y^1 = x^0$. In the convex and nonconvex settings, set $q = 0$ and $\alpha_1 = 1$.
2: **for** $s = 1, 2, \ldots, S$ **do**
3:    Find an inexact solution $x^s$ to
$$\min_x \left\{ F^s(x) := F(x) + L\|x - y^s\|^2 \right\}$$
   such that
$$\mathbb{E}\left[ F^s(x^s) - \min_x F^s(x) \right] \leq \epsilon^s.$$
4:    In the convex setting, compute $\alpha_{s+1}$ and $\beta_s$ by
$$\alpha_{s+1}^2 = (1 - \alpha_{s+1})\alpha_s^2 + q\alpha_{s+1}, \qquad \beta_s = \frac{\alpha_s(1 - \alpha_s)}{\alpha_s^2 + \alpha_{s+1}}.$$
   In the nonconvex setting, set $\beta_s = 0$.
5:    Update the prox center:
$$y^{s+1} = x^s + \beta_s(x^s - x^{s-1}).$$

6: **end for**

---

is equivalent to
$$\mathbb{E}\left[ G^s(\bar{x}^s) - \min_x G^s(x) \right] \leq \epsilon^s.$$

Moreover, the averaged prox centers satisfy
$$\bar{y}^{s+1} = \bar{x}^s + \beta_s(\bar{x}^s - \bar{x}^{s-1}).$$

Therefore, Algorithm 4 applied to the decentralized subproblems induces the standard Catalyst recursion on the averaged iterates $\{\bar{x}^s\}_{s \geq 0}$. It remains to invoke the convergence guarantee of the corresponding centralized Catalyst scheme, which we recall below.

**Theorem D.1.** *Assume that $F$ is $L$-smooth. Let $F^* = \min_x F(x)$ and $\Delta = F(x^0) - F^*$. The following guarantees hold for Algorithm 5 under the specified subproblem accuracies.*

- **Convex case.** *If $F$ is convex and $\epsilon^s = \frac{2\Delta}{9(s+1)^{4+\gamma}}$ for some $\gamma > 0$, then*
$$\mathbb{E}F(x^S) - F^* \leq \frac{8}{(S+1)^2}\left( L\|x^0 - x^*\|^2 + \frac{4\Delta}{\gamma^2} \right).$$

- **Nonconvex case.** *If $F$ is possibly nonconvex and $\epsilon^s = \frac{\Delta}{S}$, then*
$$\frac{1}{S}\sum_{s=1}^S \mathbb{E}\|\nabla F(x^s)\|^2 \leq \frac{32L\Delta}{S}.$$

*Proof.* The strongly convex and convex guarantees follow from Proposition 5 of (Lin et al., 2018), where the same argument applies under the expected subproblem accuracy condition used here.

It remains to prove the nonconvex guarantee, which follows the standard proximal-point argument for smooth nonconvex optimization; see, e.g., Drusvyatskiy & Paquette (2019). In this case $\beta_s = 0$, and hence $y^s = x^{s-1}$. Define
$$z_s := \arg\min_x \left\{ F(x) + L\|x - x^{s-1}\|^2 \right\}.$$

Since $F$ is $L$-smooth, the function $x \mapsto F(x) + L\|x - x^{s-1}\|^2$ is $L$-strongly convex. Therefore, the inexactness condition gives
$$\mathbb{E}\|x^s - z_s\|^2 \leq \frac{2\epsilon^s}{L}.$$

Moreover, by strong convexity and the same inexactness condition,

$$\frac{L}{2}\mathbb{E}\|z_s - x^{s-1}\|^2 \le \mathbb{E}\big[F(x^{s-1}) - F(x^s) + \epsilon^s\big].$$

Using the optimality condition $\nabla F(z_s) + 2L(z_s - x^{s-1}) = 0$, together with $L$-smoothness, we obtain

$$
\begin{aligned}
\mathbb{E}\|\nabla F(x^s)\|^2 &= \mathbb{E}\big\|\nabla F(x^s) - \nabla F(z_s) - 2L(z_s - x^{s-1})\big\|^2 \\
&\le 3L^2\mathbb{E}\|x^s - z_s\|^2 + 6L^2\mathbb{E}\|z_s - x^{s-1}\|^2 \\
&\le 12L\,\mathbb{E}\big[F(x^{s-1}) - F(x^s)\big] + 18L\epsilon^s.
\end{aligned}
$$

Summing this inequality over $s = 1, \ldots, S$ and using $\sum_{s=1}^S \epsilon^s = \Delta$ yields

$$\frac{1}{S}\sum_{s=1}^S \mathbb{E}\|\nabla F(x^s)\|^2 \le \frac{12L\Delta + 18L\Delta}{S} \le \frac{32L\Delta}{S}.$$

This completes the proof. $\qquad\square$

We are now ready to apply the Catalyst framework to decentralized optimization, using Algorithm 1 as the inner solver for each regularized subproblem.

**Proof of Corollary 2.5.**

*Proof.* For each outer iteration $s$, the Catalyst subproblem (72) is strongly convex with condition number $\Theta(1)$, since the proximal term contributes curvature of order $L$. Therefore, by Theorem C.2, if sufficiently many local steps are used, Algorithm 1 solves the $s$-th subproblem to accuracy

$$\mathbb{E}\big[F^s(\bar{x}^s) - \min_x F^s(x)\big] \le \epsilon^s, \qquad \bar{x}^s = \frac{1}{M}\sum_{m=1}^M x_m^s,$$

within $T_s = \widetilde{\mathcal{O}}\Big(\frac{1}{\sqrt{1-\sigma_2}}\Big)$ communication rounds.

**Convex case.** By Theorem D.1, choosing $\epsilon^s = \frac{2\Delta}{9(s+1)^{4+\gamma}}$ with $\gamma > 0$ yields

$$\mathbb{E}\big[F(\bar{x}^S) - F^*\big] \le \frac{8}{(S+1)^2}\left(L\|x^0 - x^*\|^2 + \frac{4\Delta}{\gamma^2}\right).$$

Thus, to obtain $\mathbb{E}[F(\bar{x}^S) - F^*] \le \epsilon$, it suffices to take

$$S = \widetilde{\mathcal{O}}\left(\sqrt{\frac{L\|x^0 - x^*\|^2 + \Delta/\gamma^2}{\epsilon}}\right).$$

Since each Catalyst iteration requires $\widetilde{\mathcal{O}}(1/\sqrt{1-\sigma_2})$ communication rounds, the total communication complexity is

$$\sum_{s=1}^S T_s = \widetilde{\mathcal{O}}\left(\sqrt{\frac{L\|x^0 - x^*\|^2 + \Delta/\gamma^2}{(1-\sigma_2)\epsilon}}\right).$$

Suppressing problem-dependent constants, this is $\widetilde{\mathcal{O}}(1/\sqrt{(1-\sigma_2)\epsilon})$.

**Nonconvex case.** By Theorem D.1, if we choose $\epsilon^s = \Delta/S$, then

$$\frac{1}{S}\sum_{s=1}^S \mathbb{E}\|\nabla F(\bar{x}^s)\|^2 \le \frac{32L\Delta}{S}.$$

Therefore, to find an output $\bar{x}$ satisfying $\mathbb{E}\|\nabla F(\bar{x})\|^2 \le \epsilon^2$, it suffices to take

$$S = \widetilde{\mathcal{O}}\left(\frac{L\Delta}{\epsilon^2}\right),$$

where $\bar{x}$ is chosen uniformly from $\{\bar{x}^s\}_{s=1}^S$ or as the best iterate. Equivalently, this choice gives $\epsilon^s = \Delta/S = \mathcal{O}(\epsilon^2/L)$.

For each subproblem, Algorithm 1 requires

$$T_s = \widetilde{\mathcal{O}}\left(\frac{1}{\sqrt{1-\sigma_2}}\right)$$

communication rounds. Hence the total communication complexity is

$$\sum_{s=1}^S T_s = \widetilde{\mathcal{O}}\left(\frac{L\Delta}{\sqrt{1-\sigma_2}\,\epsilon^2}\right).$$

Suppressing problem-dependent constants, this is $\widetilde{\mathcal{O}}(1/(\sqrt{1-\sigma_2}\,\epsilon^2))$. $\qquad\square$

# E. Additional Experiments

In this section, we provide additional experiments to further evaluate the empirical behavior of Local ADA. We consider three complementary settings: the effect of the momentum parameter $\beta$, an $\ell_1$-regularized quadratic problem with nonsmooth regularization, and a nonconvex MLP trained on MNIST. These experiments are intended to illustrate the robustness of Local ADA beyond the main strongly convex smooth benchmarks.

## E.1. Quadratic Problems

We first consider a decentralized strongly convex quadratic problem, where each client $m \in [M]$ has the local objective

$$f_m(x) = \frac{1}{2}(x - c_m)^\top Q(x - c_m),$$

with client-dependent vectors $c_m \in \mathbb{R}^d$ to induce data heterogeneity across agents.

**Effect of the momentum parameter.**  We study how the momentum parameter $\beta$ affects the performance of Local ADA under different numbers of local computation steps $K$. Figure 4 reports the relative error of different choices of $\beta$, which is defined as the sum of the objective gap and the consensus error, minus the corresponding value obtained when $\beta = 0$. The results show that the effect of momentum depends on the number of local steps. When $K$ is small, a large momentum parameter may hurt performance and lead to a larger relative error. In contrast, when $K$ is sufficiently large, positive momentum becomes beneficial: larger values of $\beta$ reduce the relative error compared with the no-momentum baseline. This supports the role of the momentum term in Local ADA, especially in regimes where more local computation is performed between communication rounds.

**$\ell_1$-regularized quadratic problem.**  We next consider a nonsmooth composite objective by adding an $\ell_1$ regularizer $\lambda\|x\|_1$ to the quadratic loss. We compare Local ADA with LED and Local DSGD under three decentralized network topologies: ring, grid, and exponential graph. For a fair comparison, all methods use the same number of local computation steps. As shown in Figure 5, Local ADA consistently achieves the fastest convergence across all three topologies. In particular, Local ADA reaches high-accuracy solutions using substantially fewer communication rounds than LED and Local DSGD. These results indicate that Local ADA remains effective in nonsmooth composite optimization problems.

## E.2. Nonconvex MLP on MNIST

Finally, we evaluate Local ADA on a nonconvex neural-network training task by training a two-hidden-layer MLP with $\ell_2$ regularization on the MNIST dataset under a heterogeneous non-IID data partition and a grid communication topology. All methods use the same number of local steps. Figure 6 shows the test accuracy over communication rounds. Although our theoretical guarantees focus on convex settings, Local ADA remains competitive in this nonconvex experiment. Compared with LED, Local ADA achieves nearly identical test accuracy throughout training and reaches a comparable final performance. When Local-DSGD is also included, all methods eventually achieve high test accuracy, while Local ADA remains stable and competitive.

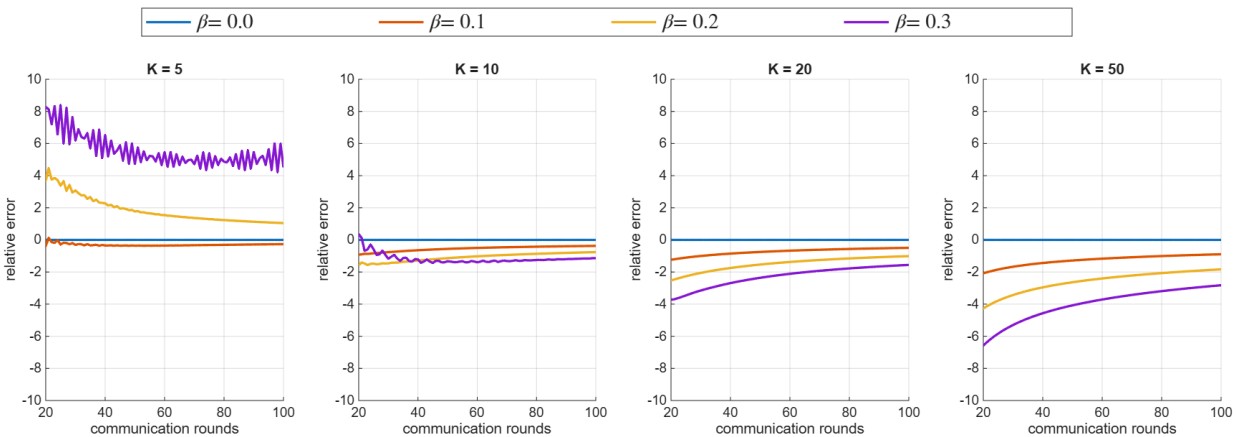

*Figure 4.* Effect of the momentum parameter $\beta$ in Local ADA under different numbers of local computation steps $K$. The $y$-axis shows the relative error, defined as the sum of the objective gap and the consensus error, minus the corresponding value when $\beta = 0$.

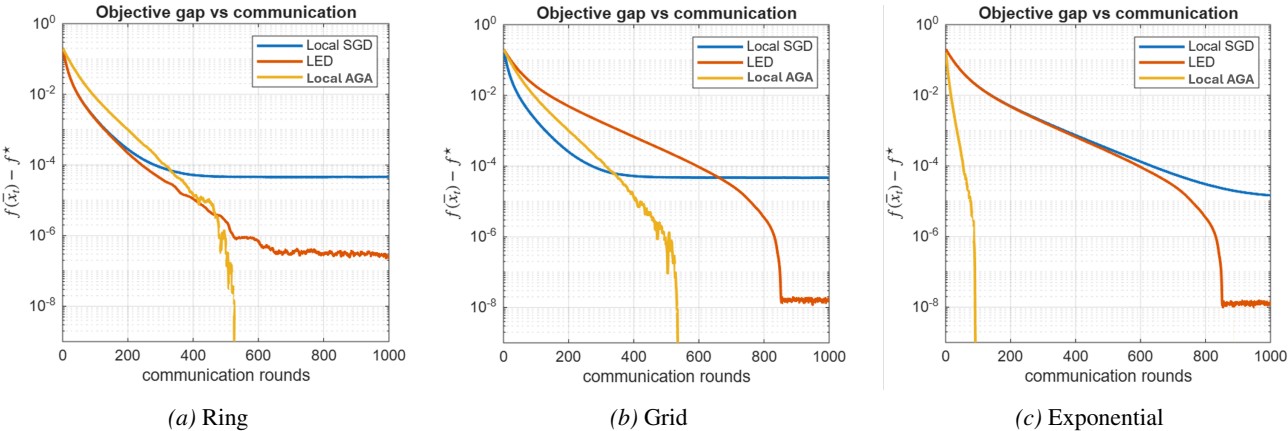

*Figure 5.* Comparison of Local ADA, LED, and Local DSGD for solving the $\ell_1$-regularized quadratic problem under three decentralized network topologies: ring, grid, and exponential graph.

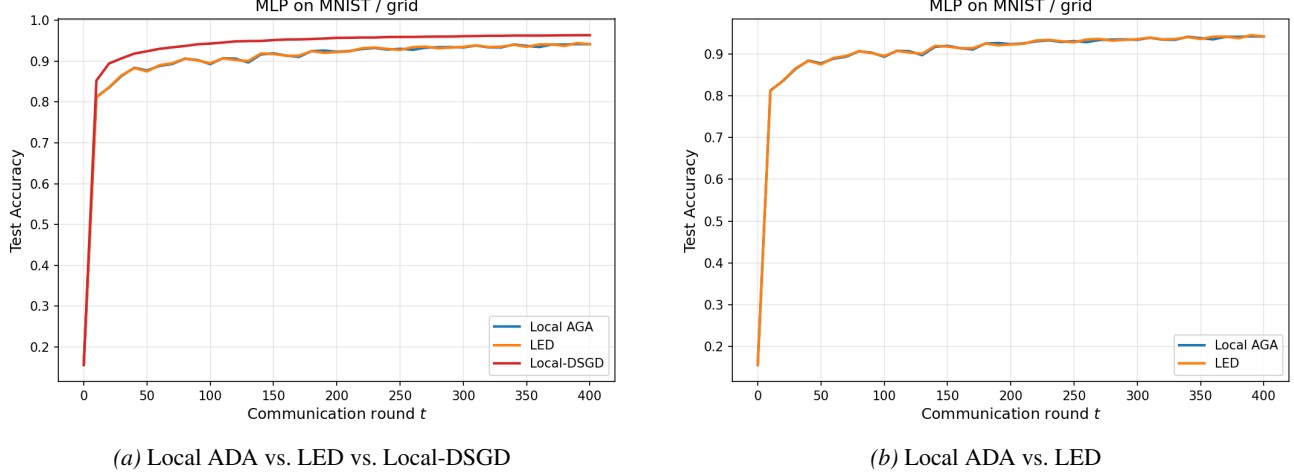

*Figure 6.* Nonconvex MLP on MNIST dataset.

