# OpenReview forum: "Accelerated Dual Method for Distributed Optimization: An Inexact-Gradient View of Local Updates"
_ICML.cc/2026/Conference — ICML 2026 regular_

### Official Review · Reviewer_iH1o · 2026-03-05

**Soundness:** 3
**Presentation:** 3
**Significance:** 2
**Originality:** 3
**Overall Recommendation:** 4
**Confidence:** 3

**Summary:**

This paper proposes accelerated dual methods for decentralized/distributed optimization that allow arbitrary numbers of local updates between communications. The method performs (accelerated) dual ascent on the consensus-constrained formulation, while each communication round computes an approximate dual gradient via K steps of local SGD on the primal variables. The key technical contribution is an inexact-gradient viewpoint: the quantity used in the dual update is treated as an inexact (and potentially biased) gradient of the dual objective, enabling a unified convergence analysis for both non-accelerated and accelerated dual schemes. The theory shows convergence for any K, and further characterizes regimes where sufficiently large K yields (near-)optimal communication complexity in the stochastic strongly convex case. Experiments on decentralized logistic regression (synthetic + w8a, multiple graph topologies) support the communication benefit of increasing K and show competitive performance versus relevant baselines.

**Compliance With Llm Reviewing Policy:**

Affirmed.

**Final Justification:**

The rebuttal addressed some of my concerns. I am fine to change my score to be 4.

**Key Questions For Authors:**

1. Composite objectives: Can the authors discuss (or sketch) how the framework might extend to smooth + nonsmooth composite objectives, potentially via proximal operators? What are the main obstacles in the current analysis?
2. Beyond strong convexity: Do any parts of the analysis extend to merely convex (or Polyak–Łojasiewicz / error-bound) settings, perhaps with different rates?
3. Parameter guidance: The method has several parameters (stepsizes, momentum, etc.). Can the authors provide more practical tuning guidance or defaults that work across graphs/datasets?

**Limitations:**

Yes.

**Strengths And Weaknesses:**

Strengths

1. Addresses an important practical question: arbitrary local steps.
   Many distributed systems choose the number of local steps based on communication/computation constraints rather than theory. Providing convergence guarantees that hold for any K is meaningful and improves robustness relative to analyses that implicitly require near-exact dual gradients.

2. Clean and general analysis via inexact gradients.
   Casting local updates as producing an inexact dual gradient gives a coherent explanation of how local computation affects global progress. The resulting framework is conceptually useful and may be reusable in other accelerated/inexact settings.

3. Strong theoretical message on communication efficiency.
   Beyond “it converges,” the paper offers a more refined picture: increasing K reduces the inexactness and can recover near-optimal communication rates under the paper’s assumptions. This helps connect theory to the common heuristic “more local work = fewer rounds.”

4. Empirical results align with the theory’s qualitative predictions.
   The logistic regression experiments show that larger K reduces required communication rounds and that the proposed accelerated variant is competitive with established baselines.

Weaknesses

1. Restrictive objective assumptions (smooth + strongly convex).
   The main results assume each local objective is differentiable, (L)-smooth, and (\mu)-strongly convex. While standard in theory, this significantly limits direct applicability to many modern ML problems, which often include nonsmooth regularizers (e.g., (\ell_1), group lasso) or composite objectives. In practice, one would want a theory that covers a smooth loss plus a nonsmooth term, or proximal variants.

2. Lack of extension to composite / nonsmooth settings.
   Given the dual viewpoint and the local-update structure, it is natural to ask whether the approach extends to objectives of the form  f_m(x)=\phi_m(x) + r(x), with (\phi_m) smooth and (r) nonsmooth but proximable. Such an extension (e.g., local proximal steps + dual acceleration) would greatly strengthen the paper’s impact.

3. Experimental scope is somewhat limited.
   Logistic regression is a reasonable benchmark, but the paper would be stronger with at least one additional task (e.g., (\ell_1)-regularized problems, or a modern federated learning benchmark). This is particularly relevant given the above limitation: demonstrating even a heuristic proximal extension empirically would help.

---

> ### Author Rebuttal · Authors · 2026-03-31
>
> Thank you for the valuable comments.
>
> > **Extension to nonsmooth composite objectives**
>
> The main difficulty in extending our framework to smooth + nonsmooth composite objectives is in Proposition 2.3: our current dual analysis relies on showing that the dual problem is both smooth and strongly concave. With an additional nonsmooth convex term, the same argument still gives smoothness of the dual, but in general only concavity, not strong concavity, which may lead to a slower rate.
> Alternatively, it may be possible to build a new primal-side analysis that can handle the composite term and still allow linear convergence in communication rounds.
>
> Following the reviewer’s suggestion, we also conducted experiments with $\ell_1$-regularized objectives on both a synthetic quadratic problem and a regression task on a real dataset. The experiment results can be found in https://anonymous.4open.science/r/ICML2026-D5D9/additional%20results.pdf. Empirically, our algorithm still exhibits fast convergence.
>
> > **Experimental scope is somewhat limited**
>
> In addition to the experiments with $\ell_1$-regularization, we added several new experiments to further validate our findings. Please refer to our response to the first reviewer (Reviewer v8qQ) for more experiment details.
>
> > **Beyond strong convexity**
>
> Our algorithm can be extended to the convex or nonconvex setting using an Inexact Accelerated Proximal Point method (APPM) or the Catalyst framework [A1, A2]., where our algorithm serves as the solver for the strongly convex subproblems. This would also give near-optimal communication complexity in these settings, i.e., $\tilde O(p^{-\frac{1}{2}}\epsilon^{-\frac{1}{2}})$ and $\tilde O(p^{-\frac{1}{2}}\epsilon^{-2})$, respectively, where $p = 1-\sigma_2$. We did not include this in the original paper because it requires an additional outer loop, and we wanted to focus on the inexact gradient framework itself.
>
> For the convex setting, another option is to add a small quadratic regularization, $O(\epsilon ||x||^2)$, to make the problem strongly convex. From the dual viewpoint in our paper, this is similar to Nesterov smoothing of the dual function.
>
> We will include this discussion in the revised paper.
>
> [A1] Cao, Tianyu, Xiaokai Chen, and Gesualdo Scutari. "Dcatalyst: A unified accelerated framework for decentralized optimization." JMLR. 2026.
>
> [A2] Li, Huan, and Zhouchen Lin. "Revisiting extra for smooth distributed optimization." SIAM Journal on Optimization. 2020.
>
>
> > **Parameter guidance for tuning**
>
> Although the result of Theorem 2.4 depend on problem parameters, it provides some guidance on their order: $\tau_1 \sim \frac{1}{T}$, $\tau_2 \sim \frac{\tau_1}{K}$, and $\beta$ (between 0 and 1) should be chosen smaller when the number of local steps is small and larger when the number of local steps is large.
> In practice, we tuned $\tau_1$ and then set $\tau_2 = \frac{\tau_1}{K}$. For $\beta$, we used 0.01 when only a few local steps were performed and 0.3 when many local steps were performed.
>
> As with most decentralized methods, some hyperparameter tuning is still needed in practice. An important future direction is to make the method tuning-free.

---

> > ### Author Rebuttal · Reviewer_iH1o · 2026-04-02
> >
> > Thank you for the additional response, but the treatment of nonsmooth and non-strongly-convex settings still relies mainly on discussion, extra experiments, and proposed extension paths (e.g., Catalyst/regularization) rather than a complete, verifiable theory with systematic comparisons, so I do not think it warrants changing my original score.

---

> > > ### Author Response · Authors · 2026-04-02
> > >
> > > We would like to clarify that extending our result to smooth convex or nonconvex settings via regularization or Catalyst is fairly standard and requires only limited additional proof.
> > >
> > >
> > > - *Convex setting with regularization:* The orignal convex smooth objective is $ F(x) = \frac{1}{M}\sum_{m=1}^M F_i(x)$. Let $x^\star$ be an optimal solution to $\min_x F(x)$, and denote $R = ||x^\star||$. Consider a regularized function
> > > $$\hat F(x) = \frac{1}{M}\sum_{m=1}^M F_i(x) + \frac{\hat \mu}{2}||x||^2 = \frac{1}{M}\sum_{m=1}^M \left\lbrace F_i(x) + \frac{\hat \mu}{2}||x||^2 \right\rbrace.$$
> > > Let $\hat F^\star$ and $F^\star$ denote the optimal values of $\hat F$ and $F$, respectively. Choosing $\hat \mu \le \frac{\epsilon}{R^2}$,
> > > $$
> > > \hat F^\star \leq \hat F(x^\star) = F(x^\star) + \frac{\hat \mu}{2}||x^\star||^2 \leq F(x^\star) + \frac{\epsilon}{2}.
> > > $$
> > > Then if $\hat x$ is a $\frac{\epsilon}{2}$-optimal solution of the regularized problem, i.e., $\hat F(\hat x) - \hat F^* \le \frac{\epsilon}{2}$, it is an $\epsilon$-solution to the original problem:
> > > $$
> > > F(\hat x) - F^\star \leq \hat F(\hat x) - F^\star = (\hat F(\hat x) - \hat F^\star ) + (\hat F^\star - F(x^\star)) \leq \frac{\epsilon}{2}+ \frac{\epsilon}{2} = \epsilon.
> > > $$
> > > Hence, it suffices to solve the regularized problem to accuracy  $\frac{\epsilon}{2}$. With the choice $\hat \mu = \frac{\epsilon}{R^2}$, the regularized $\hat F$ is $\frac{\epsilon}{R^2}$-strongly convex and $(L+ \frac{\epsilon}{R^2})$-smooth, with condition number $\hat \kappa = O(\epsilon)$. Applying our strongly convex result then gives communication complexity $\widetilde O(p^{-\frac{1}{2}}\epsilon^{-\frac{1}{2}})$, which matches the lower bound in [C1] up to logarithmic factors.
> > >
> > >
> > >
> > > - *Convex or Nonconvex setting with Catalyst:* As shown in [A2, Sec. 3.2], applying a decentralized (accelerated) proximal point algorithm (PPA) to a decentralized problem is equivalent to applying the corresponding (accelerated) PPA to the underlying centralized objective. We quickly present this argument: in each iteration $s$, decentralized (Inexact) PPA solves a decentralized problem with a strongly convex algorithm (e.g., our algorithm):
> > > $$
> > > \min_x F^s(x)=\frac{1}{M} \sum_{m=1}^M F_m^s \ \text { with } \ F_m^s(x)=F_m(x)+L||x-y_m^s||^2,
> > > $$
> > > where $y_m^s$ is is the proximal center at client $m$. Let $\bar{y}^s=\frac{1}{M} \sum_{m=1}^M y_m^s$. Then
> > > $$
> > > F^s(x)=\frac{1}{M} \sum_{m=1}^M F_m(x)+\frac{1}{M} \sum_{m=1}^M L||x-y_m^s||^2 =
> > > \frac{1}{M} \sum_{m=1}^M F_m(x)+L||x-\bar{y}^s||^2-L||\bar{y}^s||^2+\frac{L}{M} \sum_{m=1}^M||y_m^s||^2.
> > > $$
> > > Ignoring terms that does not depend on $x$, it is equivalent to
> > > $$
> > > \min_x G^s(x)=\frac{1}{M} \sum_{m=1}^M F_m(x)+L||x-\bar{y}^s||^2 = F(x) +L||x-\bar{y}^s||^2,
> > > $$
> > > which is exactly the proximal subproblem arising in the centralized PPA/Catalyst framework. Therefore, standard results for centralized PPA or Catalyst apply directly once our method is used to solve these strongly convex subproblems; see, e.g.,  [C2, C3] or [C4] (Section 4).  This yields $\widetilde O(p^{-\frac{1}{2}}\epsilon^{-2})$ communication complexity for the nonconvex setting, matching the lower bound [C5] up to logarithmic term.
> > >
> > >
> > > It is worth emphasizing that these communication complexities are attained with local steps, which have not been established in previous work. We will add these extensions as corollaries in the revised version.
> > >
> > >
> > >
> > >
> > > [A2] Li, Huan, and Zhouchen Lin. "Revisiting extra for smooth distributed optimization." SIAM Journal on Optimization. 2020.
> > >
> > > [C1] Scaman, Kevin, Francis Bach, Sébastien Bubeck, Yin Tat Lee, and Laurent Massoulié. "Optimal convergence rates for convex distributed optimization in networks." JMLR. 2019.
> > >
> > > [C2]Paquette, Courtney, Hongzhou Lin, Dmitriy Drusvyatskiy, Julien Mairal, and Zaid Harchaoui. "Catalyst acceleration for gradient-based non-convex optimization." arXiv:1703.10993. 2017.
> > >
> > > [C3] Drusvyatskiy, Dmitriy. "The proximal point method revisited." arXiv:1712.06038. 2017.
> > >
> > > [C4] Thekumparampil, Kiran K., Prateek Jain, Praneeth Netrapalli, and Sewoong Oh. "Efficient algorithms for smooth minimax optimization." NeurIPS. 2019.
> > >
> > > [C5] Lu, Yucheng, and Christopher De Sa. "Optimal complexity in decentralized training." ICML. 2021.

---

### Official Review · Reviewer_wk6t · 2026-03-11

**Soundness:** 2
**Presentation:** 2
**Significance:** 2
**Originality:** 2
**Overall Recommendation:** 4
**Confidence:** 3

**Summary:**

The paper proposes a decentralized optimization method for federated learning in which each worker performs $K$ local SGD steps on its local objective with an additional dual-driven correction term. In each synchronization round, workers synchronize their current model parameters with their neighbors via a mixing matrix, then update the dual variables. This dual update combines a gradient-ascent step with Nesterov acceleration, with the dual variables designed to reduce drift among neighboring workers and improve communication efficiency. The paper provides a convergence analysis in the stochastic strongly convex setting under heterogeneity and further develops an analysis for the convex setting with regularization to obtain reproducibility guarantees. The method is also evaluated empirically on decentralized binary classification tasks using a synthetic heterogeneous dataset and the w8a dataset.

**Compliance With Llm Reviewing Policy:**

Affirmed.

**Final Justification:**

I am satisfied that my main concerns have been addressed. The paper's primary strengths lie in improving the optimal communication complexity in a challenging, relevant decentralized learning setting with local updates. However, I also agree with Reviewer QMqr that the results are limited to the narrow case of strongly convex functions.

**Key Questions For Authors:**

Please see the weaknesses above.

**Strengths And Weaknesses:**

**Strengths**
* The paper tackles a challenging and practically important problem: analyzing decentralized learning with local update steps.
* I also find the idea of incorporating Nesterov acceleration into the dual update to be an interesting way to enhance communication efficiency.

**Weaknesses**
* The paper motivates the method in heterogeneous settings; however, the main theoretical bounds do not seem to include an explicit measure of heterogeneity. At the same time, lower-bound results in [1] suggest that some dependence on heterogeneity may be unavoidable. It would be helpful if the authors could clarify under which assumptions heterogeneity does (or does not) enter their convergence rates, and how this relates to the lower bound in [1].
* The paper addresses the stochastic setting. In distributed stochastic optimization, one typically expects improved scaling with the number of workers (for example, variance reduction proportional to $1/M$ under standard assumptions, as presented in [1]). After reviewing the appendix, I could not find this reduction at $M$ in the final complexity result. Could the authors clarify this?
* The constants $c_0$ and $c_1$ in Theorem 2.4 are not explicitly specified, which makes it difficult to evaluate the theoretical result. Could the authors provide a more explicit iteration complexity, like in [1]?
* The empirical evaluation is currently limited to relatively simple binary classification tasks. Including experiments on additional benchmarks would further strengthen the empirical validation of the proposed method.

---

[1] Koloskova, Anastasia, et al. "A unified theory of decentralized SGD with changing topology and local updates." International conference on machine learning. PMLR, 2020.

---

> ### Author Rebuttal · Authors · 2026-03-31
>
> Thanks for the helpful comments.
>
>
> > **Compare with the lower bound in [Koloskova et al., 2020]**
>
> We thank the reviewer for this question. The lower bound in [Koloskova et al., 2020] does not directly apply to our method, because our algorithm does not belong to the class considered there. In particular, their lower bound (Theorem 3) is established for algorithms within their Algorithm 1 framework, which covers a class of variants of (local) stochastic gradient descent, where each client performs an update of the form
> $$x_i^{t+1} = x_i^t - \eta g_i^t,$$
> where $g_i^t$ is a local stochastic gradient. In contrast, our method uses updates of the form
> $$x_{t, k+1}^m=x_{t, k}^m-\tau_2\left[\frac{1}{M} g_m\left(x_{t, k}^m ; \xi_k^m\right)+\tilde{\zeta}_t^m\right]$$
> which include an additional dual correction term $\tilde \zeta$. Therefore, our method falls outside their algorithm family.
>
> We also note that, even within the framework of [Koloskova et al., 2020], whether heterogeneity appears in the lower bound is tied to what problem parameters are included in $\Omega(\cdot)$. In their proof, the hard instance has heterogeneity level equal to the initialization $||x_0||^2$, and the lower bound is stated in terms of heterogeneity after replacing the dependence on the initialization by that quantity. If the dependence on $||x_0||^2$ were kept explicitly, then heterogeneity term may not be necessary in the $\Omega(\cdot)$.
>
>
> > **Scaling with the number of workers**
>
> The current result does not show a reduction in $M$. We believe this is inherent to the dual-based analysis: to obtain an estimate for the dual function/gradient, all clients need to approach their local solutions $x_i^*(\lambda)$.  This is different from a primal analysis, where averaging the primal iterates across clients reduces the variance.
>
> We would also like to note that, although local methods such as LED [1] can exhibit per-client sample complexity that improves with $M$ in a certain regime, attaining their best communication complexity still requires each client to perform $O(\epsilon^{-1})$ local steps, which does not scale down with $M$.
>
>
>
> > **Explicit constants $c_0$ and $c_1$ in the theorem**
>
> We agree that making the dependence in Theorem 2.4 more explicit would improve clarity. In particular, letting $p = 1-\sigma_2$ and $\kappa$ being condition number, we show that when $K \geq \widetilde{\Theta}\left(\max \left\lbrace \frac{c_0 \sigma^2}{\epsilon}, \kappa \right\rbrace\right) = \widetilde{\Theta}\left(\max \left\lbrace \frac{\kappa^4 \sigma^2}{\mu^2 p \epsilon}, \kappa \right\rbrace\right)$, our method attains the optimal communication complexity $\widetilde{\mathcal{O}}\left(\sqrt{\kappa/p}\right)$; Otherwise, we still guarantee $\widetilde{\mathcal{O}}\left(\frac{c_1}{\epsilon K} \right) = \widetilde{\mathcal{O}}\left(\max \left\lbrace\frac{\kappa^4 \sigma^2}{\mu^2 p^3(\epsilon K)}, \frac{\kappa^2}{K p^2}, \frac{\kappa}{p}\right\rbrace\right)$ communication rounds.
>
> While the constants in the per-client sample complexity may not be fully optimal, our main goal is to show that the optimal communication complexity $\widetilde O(\sqrt{\kappa/p})$ can be attained with $O(\epsilon^{-1})$ local steps. Compared with previous analyses of dual methods, our result guarantees convergence for any number of local steps and shows that an $O(\epsilon^2)$-accurate dual gradient, which would require $O(\epsilon^{-2})$ local samples, is not necessary.
>
>
>
> > **Additional empirical validation**
>
> We have expanded our experiments to include validation on generated quadratic functions, regression on a larger-scale dataset, and a nonconvex objective on the MNIST dataset. The experiment results can be found in https://anonymous.4open.science/r/ICML2026-D5D9/additional%20results.pdf.
> Please refer to our response to the first reviewers (Reviewer v8qQ) for more details.

---

> > ### Author Rebuttal · Reviewer_wk6t · 2026-04-04
> >
> > Thank you for your comments. My concerns have been addressed, and I raise my score to 4.

---

### Official Review · Reviewer_QMqr · 2026-03-12

**Soundness:** 3
**Presentation:** 3
**Significance:** 3
**Originality:** 3
**Overall Recommendation:** 4
**Confidence:** 3

**Summary:**

The authors introduce a method for decentralized optimization when local losses are smooth and strongly convex. The method is an accelerated gradient ascent algorithm that evolves in the dual space, but the dual gradients are inexact in that they are obtained by running a finite number of local steps of stochastic gradient in the primal space of the associated Lagrangian. The authors derive convergence guarantees for any finite number of local steps and identify a sufficient number of local steps that lead to near-optimal communication complexity.

**Compliance With Llm Reviewing Policy:**

Affirmed.

**Final Justification:**

The authors addressed my concerns. I keep the score of 4.

**Key Questions For Authors:**

Q1. It's unclear how to use Theorem 2.4 as a practical tool to design the primal and dual step-sizes \tau_1 and \tau_2 and the momentum term \beta. Can you explain how to obtain these parameters, in practice, from Theorem 2.4 (and not by trial and error)?

Q2. In line 221 of Theorem 2.4, what is the definition of an "\epsilon-accurate solution"? From line 141, second column, it seems that it relates to the Euclidean distance to the optimal point, but the proof of Theorem 2.4 suggests that it relates to the gap in the cost function. Can you clarify?

Q3. It seems that, to design the stepsizes, agents need to know the spectral gap p of the mixing matrix (defined in line 163). Is that correct? If so, how one could get this parameter in a distributed way?

**Limitations:**

Yes.

**Strengths And Weaknesses:**

Strengths:
- The authors present a meticulous theoretical analysis of the convergence properties of the proposed algorithm, which is quite challenging.

Weaknesses:
- The class of functions within the scope of the algorithm is rather limited, as all local functions are required to be smooth and strongly convex;
- Some symbols/terms are not always properly defined in their context, e.g., line 220, symbol \kappa; symbol \delta in Theorem B.1, and some typos persist, e.g., in Assumption 2.2 (1) the gradients of F_m should be Lipschitz continuous, not the functions themselves (same applies in Proposition A.1).

---

> ### Author Rebuttal · Authors · 2026-03-31
>
> Thanks for the helpful comments.
>
> > **The class of functions within the scope of the algorithm is rather limited**
>
> The paper focuses on the strongly convex setting. Our algorithm can be directly extended to the convex or nonconvex smooth setting by using an inexact accelerated proximal point method (APPM) such as Catalyst [A1, A2], where our method can solve the strongly convex subproblems generated by APPM. This would also give near-optimal communication complexity in these settings, i.e., $\tilde O(p^{-\frac{1}{2}}\epsilon^{-\frac{1}{2}})$ and $\tilde O(p^{-\frac{1}{2}}\epsilon^{-2})$, respectively, where $p = 1-\sigma_2$.
>
> We did not include this extension in the paper because it introduces an additional outer loop. We therefore focus on the strongly convex setting, where we can directly obtain the optimal communication complexity with a local-step method. We will include more discussion in our revised paper.
>
> In addition, if the local functions are nonsmooth, then the dual problem becomes smooth but only convex, not strongly convex. Extending our results to this case would require analyzing an inexact accelerated gradient method for the non-strongly convex setting. We believe our analysis in the strongly convex case paves the way for this direction.
>
> [A1] Cao, Tianyu, Xiaokai Chen, and Gesualdo Scutari. "Dcatalyst: A unified accelerated framework for decentralized optimization." JMLR. 2026.
>
> [A2] Li, Huan, and Zhouchen Lin. "Revisiting extra for smooth distributed optimization." SIAM Journal on Optimization. 2020.
>
> > **Some symbols/terms are not always properly defined in their context**
>
> Thanks for pointing them out. We have fixed them in the revised paper.
>
>
> > **How to practically choose stepsizes and momentum**
>
> Although the exact values of the parameters depend on problem-specific constants, Theorem 2.4 provides practical guidance on their order: $\tau_1 \sim \frac{1}{T}$, $\tau_2 \sim \frac{\tau_1}{K}$, and $\beta$ (between 0 and 1) should be chosen smaller when the number of local steps is small and larger when the number of local steps is large.
> In practice, we tuned $\tau_1$ and then set $\tau_2 = \frac{\tau_1}{K}$. For $\beta$, we used 0.01 when only a few local steps were performed and 0.3 when many local steps were performed.
>
> As with most decentralized methods, some hyperparameter tuning is still needed in practice. An important future direction is to make the method more tuning-free, and we will add a clarification on this point in the revision.
>
>
> > **Inconsistent definitions of $\epsilon$-accurate solution**
>
> Thank you for pointing it out. In the revised version, we will unify the definition of the optimality measure to be the objective function gap: $\mathbb{E} f(x) - f^\star \leq \epsilon$.
>
>
> > **Need to know the spectral gap p**
>
> We thank the reviewer for this question. Yes, the theoretical step-size choice depends on the spectral gap $p$. (a) If the network belongs to a standard graph family, the spectral gap may be known. (b) Otherwise, it can be estimated efficiently in a distributed way via standard procedures such as distributed power iteration [Kempe and McSherry, 2004] or other consensus-based eigenvalue estimation methods. (c) In practice, one may also tune the step-sizes and momentum empirically, as described above.
>
> [B1] Kempe, David, and Frank McSherry. "A decentralized algorithm for spectral analysis." Proceedings of the thirty-sixth annual ACM Symposium on Theory of Computing. 2004.

---

> > ### Author Rebuttal · Reviewer_QMqr · 2026-04-02
> >
> > The authors have addressed my questions. I keep the positive score.

---

### Official Review · Reviewer_v8qQ · 2026-03-13

**Soundness:** 3
**Presentation:** 3
**Significance:** 3
**Originality:** 3
**Overall Recommendation:** 4
**Confidence:** 3

**Summary:**

This paper proposes Local ADA, a primal-dual method for stochastic strongly convex distributed optimization with heterogeneous data. The main idea is to combine local multi-step SGD on primal variables with accelerated dual updates, and to analyze the dual update through an inexact-gradient framework. The paper proves convergence for arbitrary local step numbers K, and shows near-optimal communication complexity when K is sufficiently large. It also provides an additional theoretical result on reproducibility under biased gradient estimates.

**Compliance With Llm Reviewing Policy:**

Affirmed.

**Final Justification:**

Although the idea novelty is somewhat limited, the paper is overall well executed and complete. The technical development is solid, the presentation is generally clear, and the empirical study is reasonably thorough. Considering the completeness of the work, I will keep a positive score.

**Key Questions For Authors:**

1.Could the authors provide additional practical experiments, for example on larger-scale learning tasks, to better validate the empirical relevance of the method?

2.The paper highlights support for arbitrary K, but the near-optimal communication complexity seems to require K to be sufficiently large. Could the authors clarify whether the “arbitrary K” claim should mainly be interpreted as a convergence guarantee rather than an efficiency guarantee?

3.Since the current analysis depends strongly on smoothness and strong convexity, could the authors comment on whether the framework may be extendable to general convex or nonconvex settings, and what the main obstacles would be?

**Limitations:**

Yes

**Strengths And Weaknesses:**

Strengths

1.The paper presents a clean theoretical view of local updates through an inexact accelerated dual formulation. It provides convergence guarantees for arbitrary local step numbers K, and identifies the regime in which near-optimal communication complexity can be achieved.

2.The writing is also reasonably clear and logically consistent, and the overall argument is generally easy to follow.

3.In addition, the paper is well connected to prior work.

Weaknesses

1.The experimental evaluation is quite limited and does not sufficiently validate the practical claims.

2.While the method allows arbitrary K, the near-optimal complexity result only holds when K is sufficiently large, and this trade-off should be emphasized more clearly.

3.Finally, the theory relies on strong convexity and somewhat restrictive assumptions, which narrows the scope of applicability.

---

> ### Author Rebuttal · Authors · 2026-03-31
>
> Thanks for the valuable comments.
>
>
> > **Additional experiments**
>
> We conducted additional experiments on larger-scale datasets, considered objectives with nonsmooth regularization (as suggested by Reviewer iH1o), and studied the role of the momentum parameter $\beta$. The experiment results can be found in https://anonymous.4open.science/r/ICML2026-D5D9/additional%20results.pdf
>
> - *Quadratic Problems:* We considered a decentralized strongly convex problem in which each agent $m\in[M]$ is associated with a local objective
> $$
> f_m(x)=\frac{1}{2}(x-c_m)^\top Q(x-c_m),
> $$
> where $c_m\in\mathbb{R}^d$ are different across clients to create heterogeneity. We conduct two groups of experiments: (i) For our method, Local AGA (LAGA), we investigated the effect of the momentum parameter $\beta$ under different choices of the number of local steps $K$. The result is shown in Figure 1, where we compare the error for different values of $\beta>0$ with the baseline $\beta = 0$. We observe that larger $\beta$ performs better when more local steps are used.  (ii) We added an $\ell_1$-norm regularizer $\lambda \\|x\\|_1$, and compared LAGA with LED and Local SGD under three network topologies: ring, grid, and exponential. The results are shown in Figure 2.
>
> - *Regression on YearPredictionMSD Dataset:* We conduct regression experiments on the YearPredictionMSD dataset (515K samples, 90 features). We use $M=20$ clients over a $4\times 5$ grid graph ($\delta\approx 0.086$), and apply shard-based partitioning to induce data heterogeneity. (i) The result with $\ell_2$-regularization is shown in Figure 3. (ii) The result with combined $\ell_1$ and $\ell_2$-regularization is shown in Figure 4.
>
> - *Nonconvex problem with MNIST Dataset:* we train a two-hidden-layer MLP with $\ell_2$ regularization on MNIST dataset under a Dirichlet partition ($\alpha=0.2$), using $K=400$ local steps. The result is shown in Figure 5.
>
>
> > **Is “arbitrary K” claim should mainly be interpreted as a convergence guarantee rather than an efficiency guarantee?**
>
> It is unavoidable that attaining the optimal communication complexity requires $K $ to be sufficiently large. The sample complexity lower bounds for stochastic strongly convex problems is $\Omega(\epsilon^{-1})$, whereas the communication complexity lower bounds $\Omega(\log \epsilon^{-1})$. Therefore, when $K$ (the samples used in between two communication rounds) is not large enough, it is impossible to achieve the optimal communication complexity.  Even when $K$ is not large enough, we have established $O(\epsilon^{-1})$ sample complexity in $\epsilon$, which is sample efficient.
>
>
>
> > **Whether the framework may be extendable to general convex or nonconvex settings**
>
> We focused on the strongly convex setting. However, our algorithm can be directly extended to the convex or nonconvex smooth setting by using an inexact accelerated proximal point method (APPM) such as Catalyst [A1, A2], where our method can solve the strongly convex subproblems generated by APPM. This would also give near-optimal communication complexity in these settings, i.e., $\tilde O(p^{-\frac{1}{2}}\epsilon^{-\frac{1}{2}})$ and $\tilde O(p^{-\frac{1}{2}}\epsilon^{-2})$, respectively, where $p = 1-\sigma_2$.  We did not include this extension in the paper because it introduces an additional outer loop.
>
> For the convex setting, another option is to add a small quadratic regularization, $O(\epsilon ||x||^2)$, to make the problem strongly convex. From the dual viewpoint in our paper, this is similar to Nesterov smoothing of the dual function, which is nonsmooth when the primal objective is not strongly convex.
>
> We will add this discussion to the revised paper.
>
> [A1] Cao, Tianyu, Xiaokai Chen, and Gesualdo Scutari. "Dcatalyst: A unified accelerated framework for decentralized optimization." JMLR. 2026.
>
> [A2] Li, Huan, and Zhouchen Lin. "Revisiting extra for smooth distributed optimization." SIAM Journal on Optimization. 2020.

---

> > ### Author Rebuttal · Reviewer_v8qQ · 2026-04-03
> >
> > My main concerns have been resolved.

---

### Decision · Program_Chairs · 2026-04-30

**Decision:**

Accept (regular)

**Comment:**

This paper is ultimately on the positive side of the acceptance threshold. The reviewers all find value in the paper, and I agree with that overall assessment.

The main concerns raised during review were: (i) the initially limited experimental validation, which the authors substantially expanded during the rebuttal phase, and (ii) the limited scope of applicability, due in particular to the smooth and strongly convex assumptions. While this second limitation is real, I believe it is reasonable to begin with the smooth + strongly convex setting: it allows for a cleaner analysis, a sharper characterization of the rates, and more convincing conclusions at this stage.

Based on the reviewers' comments and my own reading, I believe the paper merits acceptance at ICML 2026. The theoretical contribution is meaningful, and the rebuttal addressed the main empirical concern in a satisfactory way.

That said, I believe the paper should substantially improve its discussion of related work in the revised version, including both the references raised during the review discussion and the ones listed below. In particular:

- There appear to be closely related works in which the local steps are based on proximal operators (e.g., [1] and the references therein). In this context, acceleration frameworks such as Catalyst (also mentioned in the rebuttal/discussion) or A-HPE [2] seem directly relevant and should at least be discussed. In my view, the extensions and additional discussion suggested in the rebuttal should not be added without properly positioning the paper with respect to such works, e.g., [1] (and tracking citations and references of/from it).

- More broadly, there is a large literature on approximate/inexact first-order methods, and I find it somewhat surprising that the paper refers mainly to Devolder et al. (2013, 2014) and d'Aspremont (2008) while there are several works based on more closely related inexact oracle models. For instance, [3, Proposition 4] appears particularly relevant (and may even match the same model when $\epsilon_i = 0$, if I am not mistaken). Beyond [3], there are many related references, that I let the author search, including in non-strongly convex and stochastic settings.

[1] Hendrikx, H., Bach, F., & Massoulié, L. (2019). *An Accelerated Decentralized Stochastic Proximal Algorithm for Finite Sums*
[2] Monteiro, R. D., & Svaiter, B. F. (2013). *A-HPE: An Accelerated Hybrid Proximal Extragradient Method for Convex Optimization and its Implications to Second-Order Methods*
[3] Schmidt, M., Roux, N., & Bach, F. (2011). *Convergence Rates of Inexact Proximal-Gradient Methods for Convex Optimization*